# Steady state, erosional continuity, and the topography of landscapes developed in layered rocks

Matija Perne[1,2], Matthew D. Covington[1], Evan A. Thaler[1], and Joseph M. Myre[1]

[1]Department of Geosciences, University of Arkansas, Fayetteville, Arkansas, USA
[2]Jožef Stefan Institute, Ljubljana, Slovenia

*Correspondence to:* M. D. Covington, Department of Geosciences, University of Arkansas, 340 N Campus Drive, Fayetteville, AR 72701, USA. (mcoving@uark.edu)

**Abstract.**

The concept of topographic steady state has substantially informed our understanding of the relationships between landscapes, tectonics, climate, and lithology. In topographic steady state, erosion rates are equal everywhere, and steepness adjusts to enable equal erosion rates in rocks of different strengths. This conceptual model makes an implicit assumption of vertical contacts between different rock types. Here we hypothesize that landscapes in layered rocks will be driven toward a state of erosional continuity, where retreat rates on either side of a contact are equal in a direction parallel to the contact rather than in the vertical direction. For vertical contacts, erosional continuity is the same as topographic steady state, whereas for horizontal contacts it is equivalent to equal rates of horizontal retreat on either side of a rock contact. Using analytical solutions and numerical simulations, we show that erosional continuity predicts the form of flux steady state landscapes that develop in simulations with horizontally layered rocks. For stream power erosion, the nature of continuity steady state depends on the exponent, $n$, in the erosion model. For $n = 1$, the landscape cannot maintain continuity. For cases where $n \neq 1$, continuity is maintained, and steepness is a function of erodibility that is predicted by the theory. The landscape in continuity steady state can be quite different from that predicted by topographic steady state. For $n < 1$ continuity predicts that channels incising subhorizontal layers will be steeper in the weaker rock layers. For subhorizontal layered rocks with different erodibilities, continuity also predicts larger slope contrasts than in topographic steady state. Therefore, the relationship between steepness and erodibility within a sequence of layered rocks is a function of contact dip. For the subhorizontal limit, the history of layers exposed at base level also influences the steepness-erodibility relationship. If uplift rate is constant, continuity steady state is perturbed near base level, but these perturbations decay rapidly if there is a substantial contrast in erodibility. Though examples explored here utilize the stream power erosion model, continuity steady state provides a general mathematical tool that may also be useful to understand landscapes that develop by other erosion processes.

## 1 Introduction

The formation of landscapes is driven by tectonics and climate, and often profoundly influenced by lithology, the substrate on which tectonic and climate forces act to sculpt Earth's surface. Much of our interpretation of landscapes, and their relationship to climatic and tectonic forces, employs concepts of landscape equilibrium, or steady state. Though there are a variety of types

of landscape steady state (Willett and Brandon, 2002), topographic steady state, in which topography is constant over time, is perhaps most often used in the interpretation of landscapes. Understanding of steady state also enables identification of transience within the landscape. In particular, concepts of topographic steady state and transient response to changes in climate or tectonics are frequently used within studies of bedrock channel morphology.

5 Bedrock channels are of particular geomorphic interest because they span most of the topographic relief of mountainous terrains (Whipple and Tucker, 1999; Whipple, 2004), providing the pathways through which eroded material is routed to lowlands and a primary means by which the landscape is dissected and eroded. Therefore, bedrock channels exert important controls on the relief of mountain ranges and set the pace at which mountainous landscapes respond to changes in climate or tectonic forcing. Research on bedrock channels has driven new understanding concerning the coupling between mountain

10 building, climate, and erosion (Molnar and England, 1990; Anderson, 1994; Whipple et al., 1999; Willett, 1999).

The elevation profiles of bedrock channels enable analysis of landscapes for evidence of transience, contrasts in rates of tectonic uplift, or the influence of climate (Stock and Montgomery, 1999; Snyder et al., 2000; Lavé and Avouac, 2001; Kirby and Whipple, 2001; Lague, 2003; Duvall et al., 2004; Wobus et al., 2006; Crosby and Whipple, 2006; Bishop and Goldrick, 2010; DiBiase et al., 2010; Whittaker and Boulton, 2012; Schildgen et al., 2012; Allen et al., 2013; Prince and Spotila, 2013).

15 Within this analysis, erosion rates are typically assumed to scale as power law relations of drainage area and slope, as given by the stream power erosion model (Howard and Kerby, 1983; Whipple and Tucker, 1999),

$$E = KA^m S^n, \tag{1}$$

where $E$ is erosion rate, $K$ is erodibility, $A$ is upstream drainage area, $S$ is channel slope, and $m$ and $n$ are constant exponents. While the stream power model has known limitations (Lague, 2014), it remains the most frequently used tool for channel profile analysis and landscape evolution modeling. Under steady climatic and tectonic forcing, channels are typically assumed

20 to adjust toward topographic steady state (Hack, 1960; Howard, 1965; Willett and Brandon, 2002; Yanites and Tucker, 2010; Willett et al., 2014), where uplift and erosion are balanced and topography is constant with time. This framework enables interpretation and comparison of stream profiles to identify spatial contrasts in uplift rates or transient responses to changes in tectonic or climatic forcing.

Topographic steady state has also been used to explain channel response to substrate resistance, generally leading to a con-

25 clusion that channels are steeper within more erosion resistant bedrock and less steep within more erodible rocks (Hack, 1957; Moglen and Bras, 1995; Pazzaglia et al., 1998; Duvall et al., 2004). However, this result depends on an implicit assumption of vertical contacts between strata as in Fig. 1A. Strictly speaking, topographic equilibrium does not exist when channels incise layered rocks with different erodibilities and non-vertical contacts (Howard, 1988; Forte et al., 2016). In the case of non-vertical contacts, the contact positions shift horizontally as the channel incises, resulting in topographic changes as shown in

30 Fig. 1B,C. Studies of bedrock channel morphology have primarily focused on regions with active uplift, where rock layers are often deformed and tilted from horizontal. However, a substantial percentage of Earth's surface contains subhorizontal strata. Many of these settings also contain bedrock channels, with examples including the Colorado Plateau, the Ozark Plateaus, and

the Cumberland and Allegheny Plateaus. In such settings, intuition developed from assumptions of topographic equilibrium does not necessarily apply.

Forte et al. (2016) used landscape evolution models to demonstrate that erosion rates vary in space and time in potentially complex ways as landscapes incise through layered rocks with different erodibilities. These simulations also suggest that deviation from topographic equilibrium is strongest for rock layers that are horizontal. While topographic equilibrium does not hold in general in layered rocks, here we explore whether landscapes incising layered rocks develop any kind of steady state form, and whether there are regular relationships between steepness and rock erodibility. We show that such a form does exist in some cases, and that it is a type of flux steady state that can be derived from an assumption of erosional continuity across the rock contacts. We further examine how this steady state depends on the erosion model employed and on the contact dip angle, focusing on the case of subhorizontal layers.

## 2   Erosional continuity and steady state

Conceptual models of land surface response to changing rock type typically employ the concept of topographic steady state, which makes an implicit assumption of vertical contacts between the different rock types. In topographic steady state, vertical incision rates are matched in the two rock types (Fig. 1A). Considering the opposite limit, with horizontal contacts between rocks, it seems natural to think about horizontal retreat rates rather than vertical incision rates (Fig. 1B). It is plausible that a similar steady state exists where steepness in each rock type is fixed, and horizontal retreat rates are equal at the contact. This would not be a topographic steady state, but steepness would maintain a one-to-one correspondence with rock erodibility. The land surface would retreat horizontally at a fixed rate above and below the contact while undergoing continued uplift. Generalizing between these two limiting cases, we consider a possible steady state for arbitrary rock contact dip where surface erosion rates are equal in a direction paralleling the contact plane (Fig. 1C). We refer to equal retreat in the direction of the contact plane as erosional continuity. Mathematically speaking, it means that retreat rate in the direction of a contact is a continuous function across the contact.

Physical reasoning supports the idea that landscapes in layered rocks would tend toward erosional continuity. If the upper layer retreats slower than the lower layer in the direction of the contact, this produces a steep, or possibly overhanging, land surface at the contact (Fig. 2A). This steepening or undercutting will lead to faster vertical erosion in the upper layer and drive the system towards continuity (Fig. 2C). Similarly, if the upper layer retreats faster in the direction of the contact, this produces a low slope or reversed slope zone near the contact (Fig. 2B) that can also push the system toward continuity. Therefore, the same types of negative feedback mechanisms between topography and erosion that drive landscapes to topographic steady state (Willett and Brandon, 2002) can also plausibly drive landforms near a contact into a state that maintains continuity. We refer to this hypothesized type of equilibrium as continuity steady state.

There are cases in natural systems where continuity is not maintained at all times. For example, caprock waterfalls are similar to the case in Fig. 2A. However, even in this case the discontinuity cannot grow indefinitely. If the waterfall reaches a steady size then the system has once again obtained a state where continuity is maintained in a neighborhood near the contact.

Numerical landscape evolution models do not typically allow cases such as Fig. 2A-B. Therefore, numerical models are likely to maintain continuity even more rigidly than natural landscapes. While these lines of reasoning suggest that both natural systems and landscape evolution models may be driven toward erosional continuity, here we consider continuity steady state to be a hypothesis that we test against landscape evolution models. Erosional continuity makes quantitative predictions about steady state landscapes that are elucidated below and then tested against numerical landscape evolution models.

Using the constraint of erosional continuity, one can write a very general relationship between surface erosion rates and slopes at a contact between two rock types,

$$\frac{E_1}{E_2} = \frac{S_1 - S_c}{S_2 - S_c}, \tag{2}$$

where $E_i$ and $S_i$ are vertical erosion rates and slopes, respectively, and the index refers to rock types 1 and 2. $S_c$ is the slope of the rock contact and is defined as positive in the downstream direction. This relationship results from an assumption of equal retreat rate at the contact within both rock layers in a direction parallel to the rock contact plane, as illustrated in Figs. 1C and A1. A similar relationship is used by Imaizumi et al. (2015) to examine the parallel retreat of rock slopes. If we consider the more specific case of stream power erosion through a pair of weak and strong rocks, this leads to

$$\frac{K_w S_w{}^n}{K_s S_s{}^n} = \frac{S_w - S_c}{S_s - S_c}, \tag{3}$$

where $K_w$ is the erodibility of the weaker rock, $K_s$ is the erodibility of the stronger rock, $S_w = \tan\theta_w$ and $S_s = \tan\theta_s$ are the slopes of the channel bed in each rock type, and the contact slope is $S_c = -\tan\phi$ (derivation in Appendix A). Here we have assumed that erosion processes in both rock types can be expressed with the same exponent, $n$. While $n$ may vary with rock type if erosion processes are different (Whipple et al., 2000), fixed $n$ provides a useful starting point to understand erosion of layered rocks and is also the most common choice used in landscape evolution models.

The implications of the relationship in Equation 3 are most easily understood by examining two limiting cases, a vertical contact limit, which applies whenever contact dip is large compared to channel slope, and a subhorizontal limit, which applies when contact dip is small compared to channel slope. When the contact slope is much larger than the channel slopes ($|S_c| \gg S_w, S_s$) the right hand side of Eq. (3) is approximately one, and vertical erosion rates in both rock types are roughly equal. Rock uplift can thus be balanced by erosion in both segments, and the standard relationship between channel slopes in the two rock types, normally derived from topographic equilibrium, is recovered, with

$$K_w S_w{}^n = K_s S_s{}^n. \tag{4}$$

If the contact slope is in this steep limit, but not vertical, the contact position and topography will gradually shift horizontally with erosion and vertically with uplift, while still obeying this relation derived from topographic equilibrium.

For the subhorizontal limit, where channel slopes are much greater than the slope of the contact ($S_w, S_s \gg |S_c|$), Eq. (3) simplifies to

$$K_w S_w{}^{n-1} = K_s S_s{}^{n-1} \qquad \text{or} \qquad \frac{S_w}{S_s} = \left(\frac{K_w}{K_s}\right)^{\frac{1}{1-n}}. \tag{5}$$

In this case, continuity results in roughly the same rate of horizontal retreat in both rocks at the contact, as in Fig. 1B. This contrasts with the standard assumption of equal rates of vertical erosion, and leads to unexpected behavior. Specifically, if $n < 1$, since $K_w > K_s$, higher slopes are predicted in weaker rocks, which is in strong contrast to intuition developed from the perspective of topographic equilibrium. This results because the rate of horizontal retreat within a given rock layer ($dx/dt \propto K_i S_i^{n-1}$) is a decreasing function of slope if $n < 1$. Steeper slopes can retreat more slowly horizontally because a given increment of vertical incision produces less horizontal retreat on a steeper slope than a shallower slope. For $n < 1$ vertical erosion does not increase quickly enough with slope to offset this effect. Since horizontal retreat rate is an increasing function of erodibility, continuity requires that increases in erodibility are offset by increases in slope. For subhorizontal contacts with $n > 1$, higher slopes are once again predicted in stronger rocks.

The slope ratio ($S_w/S_s$) is depicted for the vertical and horizontal limits in Fig. 3A as a function of $n$ for an erodibility contrast of $K_w = 2K_s$. In general, contrasts in the slopes within the two strata in the subhorizontal case (Eq. 5) are larger than would be predicted using the standard formulation for vertical contacts (Eq. 4). In subhorizontal rocks (i.e. whenever rock dip is small compared to channel slope), channel slopes may become sufficiently high or low to be driven to values outside the range of validity of the stream power model, particularly for cases of $n \approx 1$. Perhaps the most common value of $n$ used within landscape evolution models is $n = 1$, therefore it is also notable that the continuity relation for subhorizontal strata contains a singularity at $n = 1$ (Fig. 3). The slope ratio ($S_w/S_s$) diverges for $n \to 1^-$ and approaches zero for $n \to 1^+$. This suggests strong dependence of channel behavior on $n$ when $n$ is close to 1. The singularity results because for $n = 1$ the horizontal retreat rate is independent of slope and solely a function of erodibility and drainage area. Therefore the channel cannot maintain continuity by adjusting steepness.

## 3  Continuity steady state and stream profiles

The channel continuity relations above apply to channels within the neighborhood of a contact. Though there are clear long-term constraints on the relative retreat rates of any two contacts, these are not sufficient to determine an entire profile. However, we hypothesize that the continuity relation applies along entire profiles, and therefore that it can be used to describe a type of equilibrium state that develops in layered rocks. If this is correct then there is a one-to-one relationship between erodibility and steepness that is predicted by the continuity relations. Here we test this hypothesis using simulations of channel and landscape evolution in horizontally layered rock.

### 3.1  Methods for one-dimensional simulations and analysis

We solve the stream power model using a first order explicit upwind finite difference method. This method is conditionally stable, and the timestep was adjusted to produce a stable Courant-Friedrich-Lax number of $\mathrm{CFL} = 0.9$. The explicit upwind scheme has commonly been used for prior studies, though it is also known to produce smoothing of channel profiles near knickpoints (Campforts and Govers, 2015). The simulations employed 2000 spatial nodes, though we also ran a few cases with higher resolution that produced the same results. For simplicity, basin area was held fixed over time and was computed as a

function of longitudinal distance, with

$$A = k_a x^h, \tag{6}$$

where $k_a = 6.69 \, \text{m}^{0.33}$ and $h = 1.67$. These parameter values are representative of natural drainage networks (Hack, 1957; Whipple and Tucker, 1999). Simulations were run with $n = 2/3$, $n = 1$, and $n = 3/2$. The value of $m$ in the stream power model was adjusted according to the choice of $n$ to assure that the concavity $m/n = 0.5$, which is typical of natural channels (Snyder et al., 2000). Both high uplift ($2.5 \, \text{mm y}^{-1}$) and low uplift ($0.25 \, \text{mm y}^{-1}$) cases were run. Simulation parameters were adjusted to provide a similar number of rock contacts in each case. For the high uplift cases, rock layers were 50 m thick, whereas for the low uplift cases rock layers were 10 m thick. Longitudinal distances were also adjusted with the high uplift cases simulating 50 km long profiles and the low uplift cases simulating 200 km long profiles. Specific parameter values are provided in Table 1.

Simulation results are most easily visualized in $\chi$ space (Perron and Royden, 2013; Royden and Taylor Perron, 2013), where the horizontal coordinate $x$ is replaced with a transformed coordinate $\chi$:

$$\chi = \int_{x_0}^{x} \left( \frac{A_0}{A(x)} \right)^{m/n} \mathrm{d}x. \tag{7}$$

One advantage of this transformation is that the effect of basin area is removed such that equilibrium channels that evolve according to the stream power model appear as straight lines in this transform space. The relation predicted by Eq. (5) is invariant under the transformation to $\chi$ space, and therefore the relation also holds if slope is replaced with steepness (gradient in $\chi$-elevation space). Throughout this work, we use a value of $A_0 = 1 \, \text{m}^2$ in the $\chi$ transforms.

### 3.2 Comparison of continuity steady state and simulated profiles

For simulations where $n \neq 1$, as hypothesized, channel profiles far from base level approach a steady configuration, in which channel slope in $\chi$ space is a unique function of rock erodibility, and the profiles exhibit straight line segments in each rock type (Figs. 4,5). For the horizontally layered case, channel profiles evolve towards a state in which they are maintaining the same shape in $\chi$ space while retreating horizontally into the bedrock. For small changes in basin area, this is equivalent to a channel maintaining constant horizontal retreat rates. For non-horizontal rocks, profile shapes will gradually change in $\chi$ space, as the slope of the contact plane in $\chi$ space changes with basin area. Animations of the simulations depicted in Figs. 4 and 5 are provided in the online supplementary material.

For $n = 1$ there is no one-to-one relation between erodibility and steepness, and the profiles do not exhibit straight-line segments in each rock type. The $n = 1$ case produces this result because the horizontal retreat rates are independent of slope and purely a function of erodibility and basin area. Consequently, adjustments of slope cannot produce equal horizontal retreat rates along the channel. Instead, segments within weaker rocks will retreat more quickly than those within stronger rocks. This produces "stretch zones" as a channel crosses from weak to strong rocks and "consuming knickpoints" as a channel crosses from strong to weak rocks (Royden and Taylor Perron, 2013; Forte et al., 2016). The channels in the simulations ultimately

reach a steady stepped shape (Figs. 4C,5C) in which weak rock layers retreat until they intercept and undermine the contact with strong layers. Near-vertical cliffs, containing both strong and weak rocks, develop at the contact channels. These dynamics are described in more detail by Forte et al. (2016). It is important to note that channels in the $n = 1$ subhorizontal case contain reaches that are sufficiently steep to negate assumptions behind the stream power model. Additionally, the nature of such profiles in simulations may be strongly dependent upon the numerical algorithm employed as a result of numerical diffusion of sharp features (Campforts and Govers, 2015).

The continuity relation (Eq. 3) predicts a slope ratio rather than absolute values of slope in each rock type. The predicted slope ratio matches the slopes in the simulation at sufficient distances from base level. Notably, the counterintuitive prediction that profiles would be steeper in weaker rocks for $n < 1$ is confirmed by the simulations (Figs. 4A,5A). However, absolute slopes, and therefore entire profiles, can be predicted by realizing that continuity steady state is actually a type of flux steady state (Willett and Brandon, 2002), where the rate of uplift of rock into the domain is equal to the rate of removal of material by erosion. First, it must be noted that the weak and strong rocks experience different rates of vertical incision in the equilibrium state (Forte et al., 2016). However, since the shape of the landscape in $\chi$ space repeats with each pair of rock layers, the long term average incision rate must be the same at all horizontal positions on the stream profile. Furthermore, the topography is not growing or decaying over time after continuity steady state is reached, which means that the average incision rate at all positions is equal to the uplift rate, or, equivalently, that the system is in flux steady state. This conclusion that the long term average rate of vertical incision at each point along the profile is equal to the uplift rate leads to a relation for the erosion rate in a given layer,

$$E_1 = U \frac{(H_1/H_2) + (K_1/K_2)(S_1/S_2)^n}{1 + H_1/H_2},$$

(8)

where $E_1$ is the erosion rate of one rock layer, $H_i$ is the thickness of the $i$th layer measured in the vertical direction, and $U$ is the uplift rate (see derivation in Appendix B). Entire theoretical profiles can be constructed using this relationship, in combination with the stream power model and the continuity relation (Eq. 5), which provides the slope ratio. At a sufficient distance from base level, these profiles closely match the simulations in cases where $n \neq 1$ (Figs. 4A-B,5A-B), further confirming that continuity state is a type of flux steady state. In addition to describing behavior near contacts, continuity steady state also describes portions of the profile that are distant from contacts. For subhorizontal rocks this often produces a landscape that is quite different from that which would be predicted by topographic steady state (Fig. 3).

In continuity steady state the slopes in both rock types are different, in general, than the slopes that would be predicted by topographic steady state. Combining Eqns. 1, 5, and 8 gives

$$\frac{S_{1,\text{cont}}}{S_{1,\text{topo}}} = \left( \frac{H_1/H_2 + (K_1/K_2)^{1/(1-n)}}{1 + H_1/H_2} \right)^{1/n},$$

(9)

where $S_{1,\text{cont}}$ and $S_{1,\text{topo}}$ are the slopes for rock layer 1 that would be obtained under continuity steady state and topographic steady state, respectively. Setting the thicknesses equal, $H_1 = H_2$, and using an example case of $K_w = 2K_s$, we plot the ratio of continuity and topographic steady state slopes for both the weak and strong layers (Fig. 6). For $n < 1$ there is always a strong difference between the continuity and topographic steady state slopes in both rocks. For $n > 1$ the weak rock in continuity

steady state never has a slope more than a factor of two different than the slope that would be predicted by topographic steady state. For large $n$ the continuity steady state slopes of both weak and strong rock layers obtain the same slope as they would in topographic steady state. Additionally, if one layer is much thicker than the other (e.g. $H_1 \to \infty$), then the slope of this layer approaches the slope that it would have under topographic steady state.

Continuity steady state predicts that the ratios of slopes in the weak and strong layers are independent of layer thickness (Eq. 5). However, it also predicts that erosion rates and absolute slope values in both rocks are dependent on the thickness of the layers (Eqs. 8 and 9). To test this prediction, we resimulated the high uplift cases above with $n = 2/3$ and $n = 3/2$ and changed the layer thickness. For ease of comparison, the total thickness of both layers was kept equal to 100 m, but the weak layer thickness was increased to 90 m. As predicted, the continuity steady state slopes vary with relative layer thickness (Fig.
7). The thicker of the two rock layers adjusts its slope toward the slope that it would have under topographic steady state. Increasing the percentage of weak rock adjusts both slopes in such a way that it reduces the total topography (Fig. 7).

### 3.3   Dynamics of base level perturbations

Continuity steady state is perturbed near base level, because a constant rate of base level fall is imposed and continuity steady state requires vertical incision at different rates in each rock type. Despite this discrepancy between base level topographic
equilibrium and continuity steady state, theoretical profiles produced using Eq. (3) and Eq. (8) closely match the shapes of the profiles for the cases where $n$ is not one. Therefore, these perturbations decay rapidly away from base level in the simulated cases. However, a question remains as to what controls this decay length scale, and how typical the cases are that we have simulated.

In a horizontally layered rock sequence, a segment of stream profile with erosion rate equal to uplift is continuously devel-
oping at base level. The slope of this base level segment in $\chi$-space is given by

$$\frac{dz}{d\chi} = \left( \frac{U}{KA_0^m} \right)^{1/n}. \tag{10}$$

The difference between this slope and the continuity steady state slope produces a knickpoint that propagates upstream with a celerity in $\chi$ space given by

$$C = \frac{U}{dz/d\chi} = U^{(n-1)/n} K^{1/n} A_0^{m/n}, \tag{11}$$

As the knickpoint crosses into the other rock type, continuity demands that $C$ does not change, because $C$ is identical to horizontal retreat rate and continuity requires this to be equal across a horizontal contact. Since celerity is a monotonic increasing function of erodibility, knickpoints formed at base level in the stronger rock are slower than those formed in the weak rock. Therefore, the weak rock knickpoints catch up to the strong rock knickpoints, and the profile damps toward equilibrium as the two interact. Consequently, we can estimate the damping length scale as the $\chi$ distance at which the knickpoints generated in
weak rock at base level catch up to the knickpoints generated in strong rock at base level.

The strong rock knickpoint begins with a head start equal to the $\chi$ distance spanned by the strong rock segment, which we call $\chi_{s,0}$ and is given by

$$\chi_{s,0} = H_s \left( \frac{K A_0^m}{U} \right)^{1/n}.$$ (12)

The strong rock knickpoint will travel an additional distance $\chi_{s,+}$ before the weak rock knickpoint catches up, and these distances are related by

$$\frac{\chi_{s,0} + \chi_{s,+}}{C_w} = \frac{\chi_{s,+}}{C_s},$$ (13)

where $C_s$ and $C_w$ are the knickpoint celerities in the strong and weak rocks, respectively. The damping length scale, $\lambda = \chi_{s,0} + \chi_{s,+}$, is the distance from base level over which the weak rock knickpoint catches the strong one and can be solved for by combining Eqs. 11, 12, and 13, leading to

$$\lambda = H_s \left( \frac{K_s A_0^m}{U} \right)^{1/n} \left[ 1 + \left( (K_w/K_s)^{1/n} - 1 \right)^{-1} \right].$$ (14)

To generalize the damping behavior of the base level perturbations it is useful to analyze a dimensionless version of $\lambda$, which is normalized by $\chi_{s,0}$,

$$\lambda^* = \frac{\lambda}{\chi_{s,0}} = 1 + \left[ \left( \frac{K_w}{K_s} \right) - 1 \right]^{-1}.$$ (15)

It can be seen that the damping length scale is primarily a function of the relative erodibilities of the two rock types. When the contrast is large, damping occurs rapidly, whereas when the contrast is small the damping length scale is large. However, in this latter case there is also very little contrast in steepness, since the erodibilities are similar. Since $\chi_{s,0}$ is the $\chi$ length of the strong rock reach near base level at the moment that the weak layer becomes exposed, $\chi_{s,0}$ is less than but on the same order of magnitude as the profile distance spanned by a pair of weak and strong rock layers. Therefore, $\lambda^*$ can be interpreted as a conservative order of magnitude estimate of the number of pairs of weak and strong rocks that are required to produce damping. That is, if $\lambda^* \sim 1$ then damping should occur within a single pair. We show $\lambda^*$ as a function of the erodibility ratio for several choices of $n$ in Figure 8. Here it can be seen that if the erodibility ratio is greater than about two or three then $\lambda^* \lesssim 2$, or, equivalently, damping occurs for parts of the profile that are separated from base level by more than two sets of contacts between the two rock types. If the erodibility ratio is greater than about ten, then $\lambda^* \lesssim 1$, and damping occurs within a single pair of the two rock types.

To illustrate this damping behavior, we run two simulations with somewhat longer damping length scales. Both simulations have profile lengths of 500 km, uplift rates of 2.5 $\mathrm{mm\,y}^{-1}$, repeating rock layers with a 50 m thickness, and weak rock layers that have an erodibility of 1.5 times the strong rock layers. One case uses: $n = 1.2$, $m = 0.6$, and $K_s = 1.5 \times 10^{-5}$, whereas the second case uses: $n = 0.8$, $m = 0.4$, and $K_s = 1 \times 10^{-4}$. For the $n = 1.2$ case, $\lambda = 2.25$, and for the $n = 0.8$ case, $\lambda = 2.45$. Profiles are shown for these simulations in Figure 9. Fast knickpoints catch the slow knickpoints at roughly the calculated length scale (Fig. 9C,D). Note that the knickpoints we are describing here are breaks in steepness, which can be downstream

decreases or increases in steepness. The knickpoint interference can be seen as the gradual reduction in the size of a topographic equilibrium slope patch near base level that reaches zero size at approximately $\chi = \lambda$. This process is visualized more clearly in animations in the supplementary material that depict the damping length scale. Beyond this damping length scale, some minor perturbations remain, and one can see fast and slow knickpoints migrating through the upper parts of the profile as the system

evolves. However, beyond $\lambda$ the theoretical profiles derived from continuity and flux steady state are a good approximation to profile shape.

## 4   Full landscape simulations

To determine whether continuity steady state is obtained within whole landscape models, or whether addition of hillslope processes might eliminate it, *FastScape* V5 (Braun and Willett, 2013) was used to simulate stream power erosion coupled to

an entire landscape model. All simulated cases employ a constant rock uplift rate and horizontal rock layers with alternating high and low erodibility.

The stream power model used in *FastScape* has the form

$$E = K_f \Phi^m S^n, \tag{16}$$

where $\Phi$ is discharge, calculated as the product of the drainage area and the precipitation rate $P$. Each of the three presented model runs uses two different erodibility coefficients, $K_{fw}$ for the weak rock and $K_{fs}$ for the strong rock, in place of $K_f$. For

each one of them, a grid of $3000 \times 3000$ pixels representing $100 \times 100$ km is simulated. The initial condition used is a slightly randomly perturbed flat surface at base level. The boundary condition is open on all sides. 15000 m of uplift is simulated in 60000 timesteps. The weaker rock is exposed for the first 10800 m of the uplift, allowing an initial drainage network to establish. Afterwards, a layered rock structure starts to be exposed, with alternating layers of 200 m of the stronger rock and 300 m of the weaker rock. The main difference between the model runs is in the slope exponent $n$, with cases using $n = 2/3$,

$n = 1$, and $n = 3/2$. A listing of numerical parameters is provided in Table 2. The necessary timestep was calculated from the uplift rate and the ratio of total uplift to the number of timesteps.

Floating point digital elevation models (DEMs) were produced for the final time step for each *FastScape* simulation. Using the *Landlab* landscape evolution model (Tucker et al., 2013) to calculate flow routing, channel profiles were extracted from the *FastScape* DEMs for each case of $n$. *Landlab* was extended to enable calculation of $\chi$ values for each channel. $\chi$-plots were

then generated for 50 channels in each simulation and are shown in Fig. 10. The continuity equilibrium state described above is also reflected within the full landscape evolution model, and plots of elevation versus $\chi$ for channels within each model demonstrate similar relationships as displayed in Figure 4A,C,E.

## 5   Discussion

Topographic steady state is not attained within layered rocks with non-vertical contacts since the spatial distribution of erodi-

bility changes in time (Howard, 1988; Forte et al., 2016). Forte et al. (2016) show that departures from topographic steady state

are greatest when the layers have contacts that are near horizontal. They use simulations of landscape evolution with a stream power erosion model with $n = 1$. These simulations demonstrate that erosion rates vary across the landscape in complex ways, that there is no direct relationship between rock erodibility and erosion rate, and that erosion rates can be greater or less than the uplift rate. They also detect distinct differences in landscape development between cases where either the strong or weak rock is exposed first. In the case of a weak rock on top of a strong rock, a tapered wedge of weak rock forms on top of a steep retreating escarpment in the strong rock. When strong rock is on top of weak rock, the weak rock undercuts the strong rock and forms an extremely steep zone near the contact.

Our simulations and analysis support the conclusions of Forte et al. (2016) on the dynamics of the $n = 1$ case. However, we also show that these dynamics result specifically because the rate of horizontal retreat, or equivalently the knickpoint celerity, is independent of slope when $n = 1$. Consequently, the topography is unable to maintain a state of erosional continuity, and therefore topography is unable to reach continuity steady state. Landforms developed in layered rocks are driven toward continuity steady state by the same type of negative feedback mechanisms between topography and erosion that generate topographic steady state. In fact, topographic steady state is a special case of continuity steady state. For stream power erosion with $n \neq 1$, landscapes are able to adjust slope to maintain continuity across multiple rock layers. Therefore, a type of equilibrium landscape form does develop sufficiently far from base level when $n \neq 1$.

If we compare the $n \neq 1$ case with the conclusions above concerning the $n = 1$ cases, several similarities and differences emerge. For both cases, it is true that topographic steady state is only strictly reached if contacts are vertical. Also, for both cases the patterns of steepness in the landscape diverge most strongly from those predicted by topographic steady state when rocks are horizontally layered. However, for $n \neq 1$ erosion rates and steepnesses do exhibit one-to-one relationships with rock erodibility. In our simulations, we do not see any dependence of topography on the order of exposure of the layers, unlike with the $n = 1$ case. Considering two rock types, one strong and one weak, erosion rates bracket the uplift rate, with one rock exhibiting erosion rates higher than uplift and the other lower than uplift. For the subhorizontal case, the weak rock erodes faster when $n < 1$ and the strong rock erodes faster when $n > 1$ (Fig. 6). Contrasts in erosion rates become small for large $n$ (Fig. 6) and very large when $n \approx 1$.

As noted by Forte et al. (2016), variability in erosion rates across the landscape can produce bias in detrital records, as zones exhibiting faster erosion will contribute a larger proportion of the exported sediment than would be calculated based on areal estimates. Since the framework developed predicts a regular relationship between erosion rates and erodibility for $n \neq 1$, it may help constrain uncertainties in such records. The long term average erosion rate at any location is equal to uplift rate, and therefore continuity steady state is a type of flux steady state. Because of this, there is also a simple rule that emerges when considering erosion rates as a function of rock type. For the portion of the landscape that is in flux steady state, the amount of material removed from a given rock layer within a period of time will be proportional to the fraction of the topography that is spanned by that layer, as opposed to its areal extent. For example, in our simulations where each rock type makes up half of the topography, there is an approximately equal volume of material eroded from each rock type within a given timestep.

When contacts between rocks dip at slopes much greater than the channel slope, then the vertical contact limit from Eq. (4) applies and topography approaches the form that would be predicted by topographic steady state. The considerations

introduced here become important as rock dips approach values comparable to or less than channel slope. This subhorizontal limit, given by Eq. (5), is most likely to apply for rocks that are very near horizontal and/or channels that are very steep. Therefore, these considerations are most applicable in cratonic settings, in headwater channels, or when considering processes of scarp retreat in subhorizontal rocks (Howard, 1995; Ward et al., 2011). In the subhorizontal limit, slope contrasts are larger than would be predicted by topographic steady state (Fig. 3). In the case of $n < 1$, slope patterns in continuity steady state are also qualitatively different than those predicted by topographic steady state, with steeper channel segments in weaker rocks. Since the relationship between erodibility and steepness within layered rocks is a function of contact dip, this may complicate the determination of erodibility using channel profile analysis in settings where the subhorizontal limit applies.

For $n \approx 1$ slope contrasts become extreme, which is particularly important since $n = 1$ is the most common value used in landscape evolution models. In this case, large slope contrasts at contacts may accentuate numerical dispersion. It also must be realized that $n = 1$ is quite a special case in subhorizontal rocks, and the rest of the parameter range for $n$ results in substantially different dynamics and steady state. Field studies have suggested that $n = 1$, where knickpoint retreat rate is independent of slope, can explain the distribution of knickpoints within drainage basins (Crosby and Whipple, 2006; Berlin and Anderson, 2007). However, it is also clear from our analysis that with $n = 1$ in subhorizontal rocks channels near contacts obtain a steep state, where the stream power model will break down.

During constant uplift, channels cannot attain continuity steady state at base level, because it requires different vertical incision rates in each rock type. However, the perturbations introduced by stream segments in topographic equilibrium at base level rapidly decay over a length scale that is primarily a function of the ratio of rock erodibilities, with larger erodibility contrasts resulting in shorter decay lengths. Practically speaking, for rocks that have erodibilities sufficiently different to have a strong effect on the profile, base level perturbations of continuity steady state decay after a couple rock contacts are passed.

Though steepness ratios are a fixed function of rock erodibility in continuity steady state, absolute steepness values depend on rock layer thickness. Since natural systems will not generally have regular patterns of thickness or erodibility, this has implications for the ability of natural systems to approach continuity steady state. As new rock layers with different thicknesses or erodibilities are exposed at base level, the absolute steepness values that would represent continuity steady state change. Therefore, continuity steady state may often represent a moving target, where the landscape is constantly adjusting toward it but never reaching it. The introduction of rock layers with varying thickness and erodibility can produce transience in landscapes that are experiencing otherwise stable tectonic and climate forcing. This only applies, however, for absolute steepness values. Steepness ratios, and their relationship to erodibility, would be expected to be relatively constant in time if sufficiently far from base level. Since the relationship between erodibility and steepness will change in both time and space as new layers are exposed at base level, this may confound attempts to identify erodibility values using channel profiles within steep channels in subhorizontal rocks. However, since steepness ratios do not depend on these dynamics, analysis of steepness ratios derived from profiles, rather than absolute steepness values, may enable quantification of the relative erodibility of layers.

We speculate that the simulated dynamics in subhorizontal rocks provide a potential means to generate caprock waterfalls, a feature that has long fascinated geologists (Gilbert, 1895). Caprock waterfalls, such as Niagara Falls, have a resistant caprock layer that is underlain by a weaker rock. The waterfall has the caprock at its lip, followed by a vertical, or often overhanging,

face within the weak rock. This is a case of a very steep channel within a highly erodible rock, which would not be predicted from topographic equilibrium and stream power erosion. Such a state is predicted by the continuity relation developed here for subhorizontal layers with $n < 1$, and somewhat similar features develop in the case of $n = 1$. Values of $n$ might be expected to be less than one for erosion processes active in the weak rock layer, such as plucking (Whipple et al., 2000). Furthermore, caprock waterfalls typically form in relatively horizontal strata, and are common within steep headwater channels, which are the settings where differences between topographic and continuity steady state become important. The stream power model arguably does not apply to waterfalls (Lamb and Dietrich, 2009; Haviv et al., 2010; Lague, 2014), and a variety of erosion mechanisms that are independent of stream power can act in such an oversteepened reach, such as gravity failure, freeze-thaw, shrink-swell, and seepage weathering. However, starting from an initial condition of low relief, topographic equilibrium and stream power erosion would not predict a channel to evolve toward the caprock waterfall state. In contrast, the framework presented here naturally produces features resembling caprock waterfalls from considerations of landscape equilibrium. While further work would be needed to test this hypothesis, it remains plausible that caprock waterfalls are the result of channels steepened within weaker rocks to maintain continuity, even if, once the channel becomes sufficiently steep, stream power erosion no longer provides a good approximation to erosion rates. The concept of continuity could also be applied to other, more mechanistic, erosion models, as the relation provided by Eq. 2 is independent of erosion model. However, continuity relations are most likely to provide insight for simple erosion models where analytical solutions can be derived, as with stream power erosion. With more complex models, the results of numerical landscape evolution models could be compared against the continuity relation to test whether a similar continuity steady state is attained.

Though the focus of this work is on bedrock channel profiles in layered rocks, the concepts of continuity and flux steady state can be applied in general to any mathematical model for erosion. Much like topographic steady state, both continuity and flux steady state result from negative feedback within the uplift-erosion system that drive it toward steady state as uplift and erosion become balanced. Such feedback mechanisms are likely to be present within most erosional models. Though topographic steady state has been a powerful theoretical tool to understand landscapes, the generalized concept of erosional continuity may prove more useful in interpreting steep landscapes in subhorizontal rocks.

# 6 Conclusions

Topographic steady state has provided a powerful tool for understanding the response of landscapes to climate, tectonics, and lithology. However, within layered rocks, topographic steady state is only attained in the case of vertical contacts. In topographic steady state, vertical erosion rates are equal everywhere, and steepness adjusts with rock erodibility to produce equal erosion. Here we generalize this idea using the concept of erosional continuity, which is a state where retreat rates of the land surface on either side of a rock contact are equal in the direction parallel to the contact rather than in the vertical direction. Using a stream power erosion model with $n = 1$, prior work showed that erosion rates exhibit transient and complex relationships with rock erodibility (Forte et al., 2016). Our work suggests that these complex and transient effects result because adjustments in steepness cannot produce a state of erosional continuity when $n = 1$. In cases where $n \neq 1$, erosional continuity

can be attained, and the landscape sufficiently far from base level exhibits one-to-one relationships between steepness and erodibility that are predicted by continuity. We refer to this as continuity steady state, and show that it is a type of flux steady state. Results from 1D and 2D landscape evolution models confirm the predictions of the erosional continuity equations.

For continuity steady state, the relationships between rock erodibility and landscape steepness differ most from topographic steady state when the rock contacts are subhorizontal, that is, when contact dips are less than channel slope. In the subhorizontal case, contrasts in steepness are larger than predicted by topographic steady state. These contrasts are largest when $n \approx 1$, and in fact may create sufficiently steep channels in one of the rock layers to negate the applicability of the stream power erosion model. For $n \approx 1$, numerical dispersion may also influence the time evolution of the topography because of the large slope contrasts. When $n < 1$, steepness patterns are also qualitatively different than those predicted by topographic steady state, with steeper channel segments in weaker rocks. In continuity steady state, erosion rates bracket the uplift rate and display a regular relationship with erodibility. This may assist in quantifying the uncertainty and bias within detrital records that can result from different erosion rates in different rock types (Forte et al., 2016). Since relationships between erodibility and steepness are both a function of rock dip and the history of layers exposed at base level, this may confound attempts to identify erodibility values using stream profile analysis in some settings. For subhorizontal rocks, continuity steady state is not attained at base level. However, the perturbations to continuity steady state that are introduced at base level decay rapidly when there is a contrast in erodibility of more than a factor of two to three. We speculate that the framework developed here provides a possible mechanism for the development of caprock waterfalls, since it predicts steep channel reaches within weak rocks. Though we focus on stream power erosion, the concept of erosional continuity is quite general, and may provide insight when applied to other erosion models.

## Appendix A: Derivation of the continuity relation

Here we detail how the constraint of channel continuity can be used to derive the relationship given in Eq. (3). Consider a planar contact between rock types with different erodibilities. We label the downstream and upstream erodibility with $K_1$ and $K_2$. Downstream and upstream slopes are $S_1$ and $S_2$, the slope of the contact is $S_c$, and their respective slope angles are $\theta_1$, $\theta_2$, and $\phi$, see Fig. A1.

In this section we use the subscript $i$ to denote either 1 or 2, as the relationships are valid for the channel within both rock types. Erosion at a rate $E_i$ in the vertical direction, as is calculated by the stream power model, can be transformed to an erosion rate $B_i$ that is perpendicular to the channel bed using the slope of the channel bed, $\theta_i$, with $B_i = E_i \cos \theta_i$, see Fig. A1. The contact and the channel intersect at angle $\theta_i + \phi$, thus the rate of exposure of the contact plane is

$$R_i = \frac{B_i}{\sin(\theta_i + \phi)} = \frac{E_i \cos \theta_i}{\sin(\theta_i + \phi)}. \tag{A1}$$

For the case where $\theta_i + \phi > \pi/2$ the diagram changes, but these same relationships can be recovered using $\sin(\pi - \theta_i - \phi) = \sin(\theta_i + \phi)$. Continuity of the channel bed requires that the contact exposure rates $R_1$ and $R_2$ are equal, which gives

$$\frac{E_1 \cos \theta_1}{\sin(\theta_1 + \phi)} = \frac{E_2 \cos \theta_2}{\sin(\theta_2 + \phi)}. \tag{A2}$$

Using a trigonometric identity for angle sums leads to

$$\frac{E_1 \cos\theta_1}{\sin\theta_1 \cos\phi + \cos\theta_1 \sin\phi} = \frac{E_2 \cos\theta_2}{\sin\theta_2 \cos\phi + \cos\theta_2 \sin\phi}. \tag{A3}$$

Simplifying the fractions and multiplying both sides of the equation with $\cos\phi$ we get

$$\frac{E_1}{\tan\theta_1 + \tan\phi} = \frac{E_2}{\tan\theta_2 + \tan\phi}. \tag{A4}$$

Solving for the ratio of erosion rates in the two rock types and converting to slopes rather than angles, using a sign convention where both contact and bed slopes are positive in the downstream direction, the relation becomes

$$\frac{E_1}{E_2} = \frac{S_1 - S_c}{S_2 - S_c}. \tag{A5}$$

If erosion rates are given by the stream power model, then it follows that

$$\frac{K_1 S_1^n}{K_2 S_2^n} = \frac{S_1 - S_c}{S_2 - S_c}, \tag{A6}$$

which is identical to Eq. (3) with the general subscripts 1 and 2 replaced with $s$ and $w$ for strong and weak.

## Appendix B: Derivation of the erosion relation

Using the stream power model, erosion rates in two channel segments above and below a contact are

$$E_1 = K_1 A^m S_1^n \qquad \text{and} \qquad E_2 = K_2 A^m S_2^n, \tag{B1}$$

where $A$ is the recharge area. Taking the ratio of both equations at an arbitrary basin area, we get

$$\frac{E_1}{E_2} = \frac{K_1}{K_2}\left(\frac{S_1}{S_2}\right)^n. \tag{B2}$$

We define $H_1$ and $H_2$ to be the thicknesses of the rock layers measured in the vertical direction. If flux steady state is assumed, then the average erosion rate equals the uplift rate $U$. Therefore, the time needed to uplift a distance equal to the sum of the thicknesses of the two layers equals the sum of the times needed to erode through the two layers:

$$\frac{H_1 + H_2}{U} = \frac{H_1}{E_1} + \frac{H_2}{E_2}. \tag{B3}$$

Combining Eq. (B2) and Eq. (B3) gives an expression for the erosion rate in a given rock:

$$E_1 = U\frac{H_1/H_2 + K_1/K_2\left(S_1/S_2\right)^n}{1 + H_1/H_2}. \tag{B4}$$

While flux steady state seems like a reasonable assumption, simulations also confirm that the erosion rates predicted by Eq. (B4) are approached within a few contacts above base level. Similarly, simulations that alternate uplift rate over time to match the erosion rate of the rock type currently at base level, as given by Eq. (B4), obtain straight line slopes in $\chi$-elevation space

all the way to base level. This confirms that the disequilibrium seen in the profiles in Fig. 4B-D is produced by the difference between the constant uplift rate and the equilibrium incision rates experienced in each layer.

*Acknowledgements.* MP acknowledges the support of the Slovenian Research Agency through Research Programme P2-0001. MDC, MP, and EAT acknowledge support from the National Science Foundation under EAR 1226903. Any opinions, findings, conclusions or recommendations expressed in this material are those of the author and do not necessarily reflect the views of the National Science Foundation. The simulation outputs and code on which this work are based are available upon request.

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

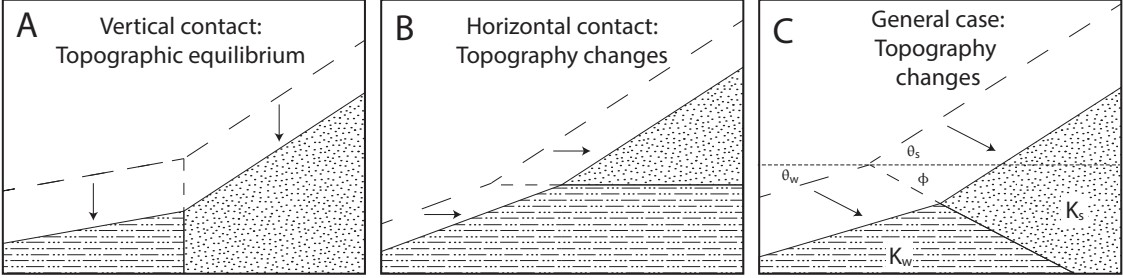

**Figure 1.** Topographic equilibrium in layered rocks. (A) Response of steepness to rock erodibility is typically derived from a perspective of topographic equilibrium, with equal vertical incision rates in all locations that are balanced by uplift. Topographic equilibrium does not occur in the case of non-vertical contacts. (B) For horizontal strata, horizontal retreat rates, rather than vertical incision, must be equal at the contact. (C) In general, retreat in the direction parallel to the contact must be equal within both rocks to maintain channel continuity. Dashed lines depict former land surface and contact positions, and arrows show the direction of equal erosion at the contact. Uplift is not depicted.

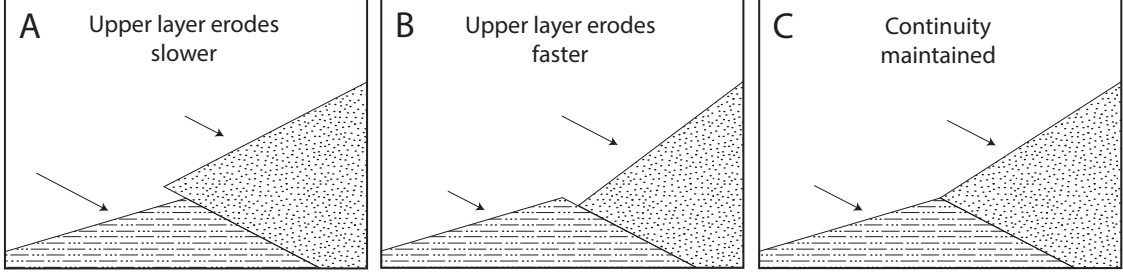

**Figure 2.** Erosional continuity. (A) If the upper layer at a contact erodes slower, this produces a discontinuity at the contact and the resulting steepening or undercutting of the upper layer will drive the system toward erosional continuity. (B) If the upper layer erodes faster, this produces a low or reversed slope zone near the contact, which will also drive the system toward continuity (C) We hypothesize that, in general, topography will tend to approach a state where continuity is maintained.

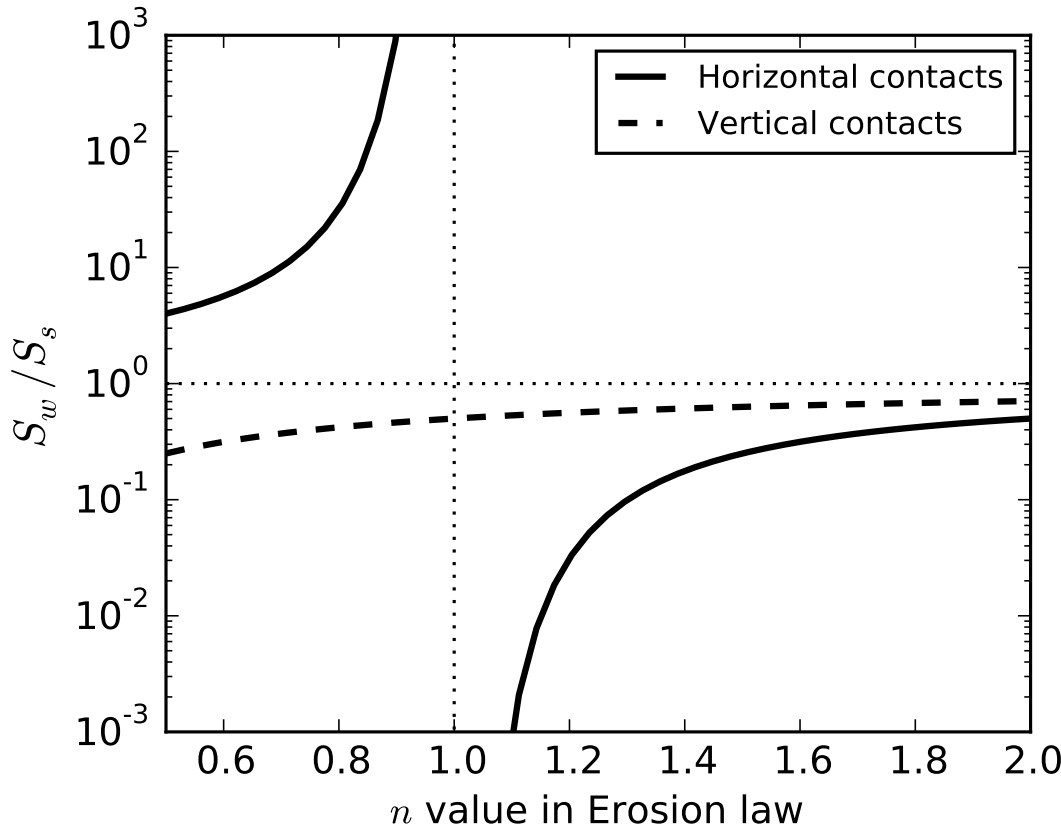

**Figure 3.** Channel slope response at a subhorizontal contact from an assumption of continuity. The ratio of slope within the weaker rock ($S_w$) and the slope within the stronger rock ($S_s$) near a horizontal contact (solid line) with differing values of the exponent $n$ in the stream power model. Erodibility in the weaker rocks ($K_w$) is twice that of the stronger rocks ($K_s$). This subhorizontal case applies when the dip of the contact is small compared to channel slope. The dashed line displays the standard topographic equilibrium relationship, which applies for cases where the contact slope is much larger than the channel slope.

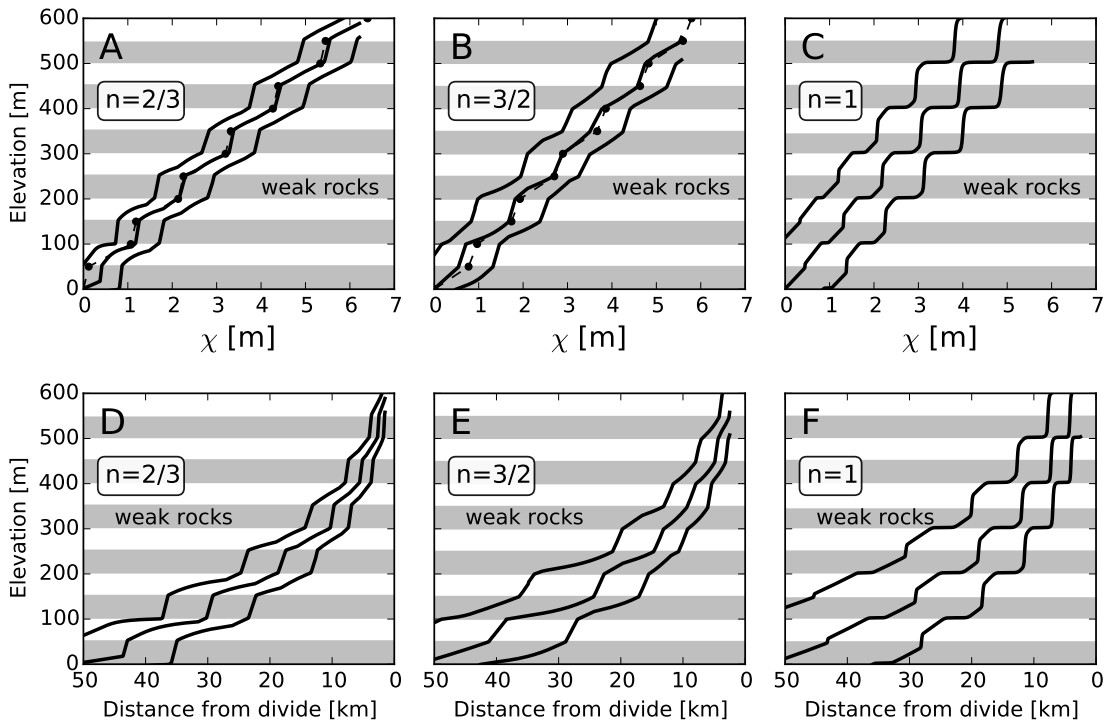

**Figure 4.** Channel profiles in subhorizontally layered rocks with high uplift ($2.5\,\mathrm{mm\,yr^{-1}}$). (A-C) Channel profiles in $\chi$-elevation space for cases where $n=2/3$ (A), $n=3/2$ (B), and $n=1$ (C). (D-F) Channel profiles as a function of distance from divide. Each panel contains three time snapshots of the profile with uplift subtracted from elevation so that the profiles evolve from left to right. Grey bands represent the weak rock layers. The dashed lines (A,B) show the profiles predicted by the continuity steady state theory (Eqs. 5 and 8), with filled circles depicting predicted crossing points of the contacts. Channel profiles obtain a steady state shape except near base level, where a constant rate of base level fall is imposed. For $n \neq 1$ the equilibrium profile steepness (slope in $\chi$ space) has a one-to-one relationship with rock erodibility, with steeper channels in weaker rock if $n < 1$. For $n = 1$ there is no unique relationship between erodibility and steepness, as continuity cannot be maintained along the entire profile.

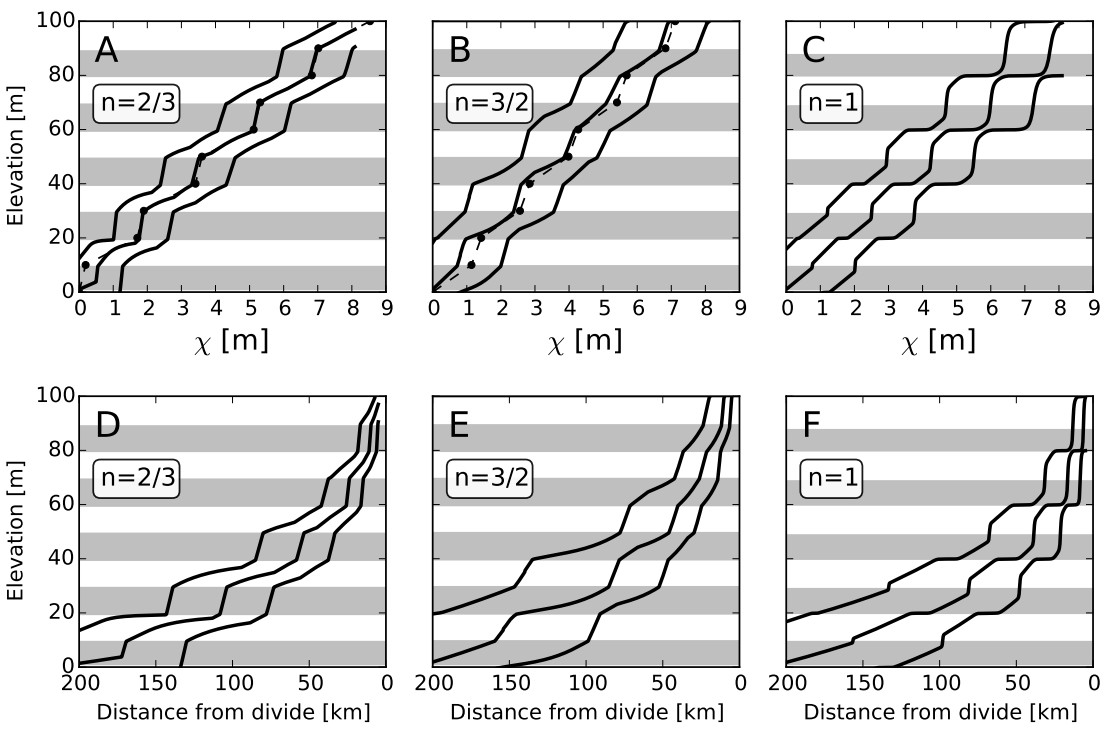

**Figure 5.** Channel profiles in subhorizontally layered rocks with low uplift ($0.25 \ \mathrm{mm \, yr^{-1}}$). (A-C) Channel profiles in $\chi$-elevation space for cases where $n = 2/3$ (A), $n = 3/2$ (B), and $n = 1$ (C). (D-F) Channel profiles as a function of distance from divide. Grey bands indicate weaker rocks. The low uplift simulations utilize longer distances and thinner rock layers in order to obtain a similar number of rock layer cycles. These profile shapes are qualitatively similar to the high uplift cases (Fig. 4).

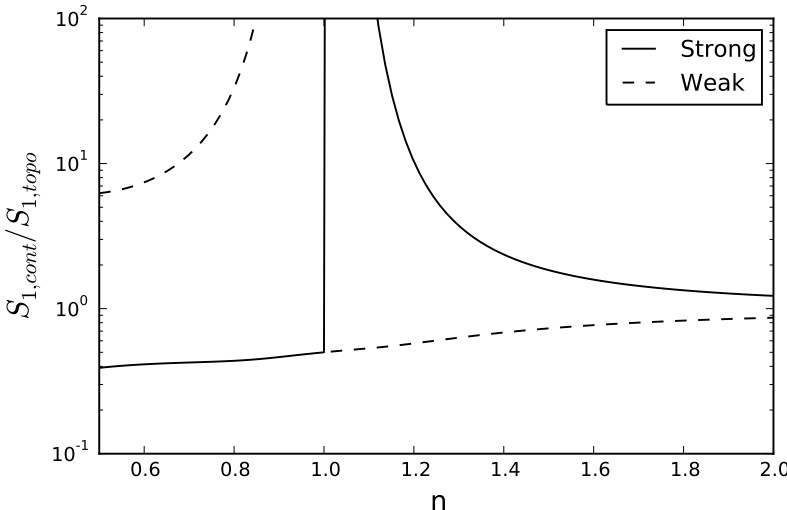

**Figure 6.** An example case of the ratio of slopes predicted by continuity and topographic steady states. This example assumes a choice of equal rock thicknesses in both rock types and a weak rock erodibility that is twice that of the strong rock. Contrasts are in general strongest for $n < 1$ and gradually disappear for large $n$.

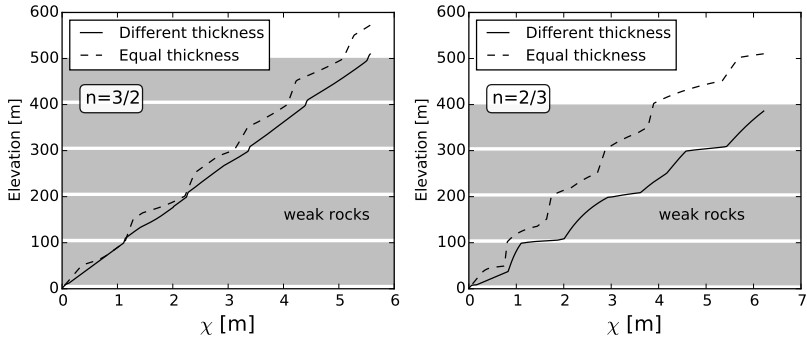

**Figure 7.** The influence of relative layer thickness on slopes in continuity steady state. If the relative thickness of the strong and weak layers is changed, the far from base level slopes in both rocks adjust correspondingly (solid lines), as predicted by continuity steady state. Grey bands depict the locations of weak rocks in the differing thickness model. The dashed lines depict channel profiles for simulations with equal layer thickness but the same erosional parameters. Increasing the weak layer percentage reduces topography overall.

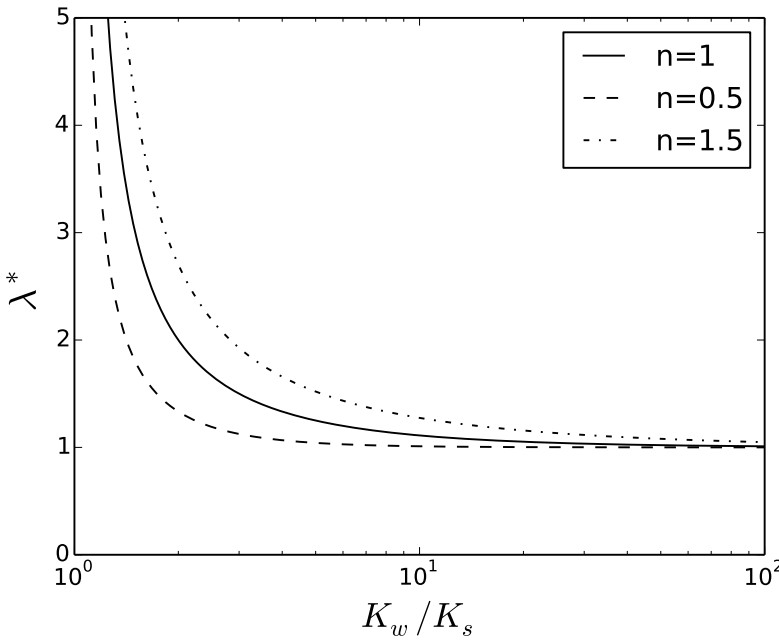

**Figure 8.** The dimensionless damping length scale, $\lambda^*$, as a function of erodibility ratio. Damping of base level perturbations is strong when the erodibility ratio is greater than three. $\lambda^*$ can be interpreted as roughly the number of pairs of strong and weak rock layers that base level perturbations must pass through before substantial damping toward continuity steady state.

| Simulation | $K_s\,[\mathrm{m}^{1-2m}\,\mathrm{a}^{-1}]$ | $K_w\,[\mathrm{m}^{1-2m}\,\mathrm{a}^{-1}]$ | $m$ | $U\,[\mathrm{m}\,\mathrm{a}^{-1}]$ |
|---|---|---|---|---|
| **High uplift cases** | | | | |
| $n=2/3$ | $1 \cdot 10^{-4}$ | $2 \cdot 10^{-4}$ | $1/3$ | $2.5 \cdot 10^{-3}$ |
| $n=1$ | $2 \cdot 10^{-5}$ | $2.4 \cdot 10^{-4}$ | $1/2$ | $2.5 \cdot 10^{-3}$ |
| $n=3/2$ | $1.5 \cdot 10^{-6}$ | $3 \cdot 10^{-6}$ | $3/4$ | $2.5 \cdot 10^{-3}$ |
| **Low uplift cases** | | | | |
| $n=2/3$ | $4 \cdot 10^{-5}$ | $8 \cdot 10^{-5}$ | $1/3$ | $2.5 \cdot 10^{-4}$ |
| $n=1$ | $2 \cdot 10^{-5}$ | $2.4 \cdot 10^{-4}$ | $1/2$ | $2.5 \cdot 10^{-4}$ |
| $n=3/2$ | $3 \cdot 10^{-6}$ | $6 \cdot 10^{-6}$ | $3/4$ | $2.5 \cdot 10^{-4}$ |

**Table 1.** Parameters used in the 1D model runs.

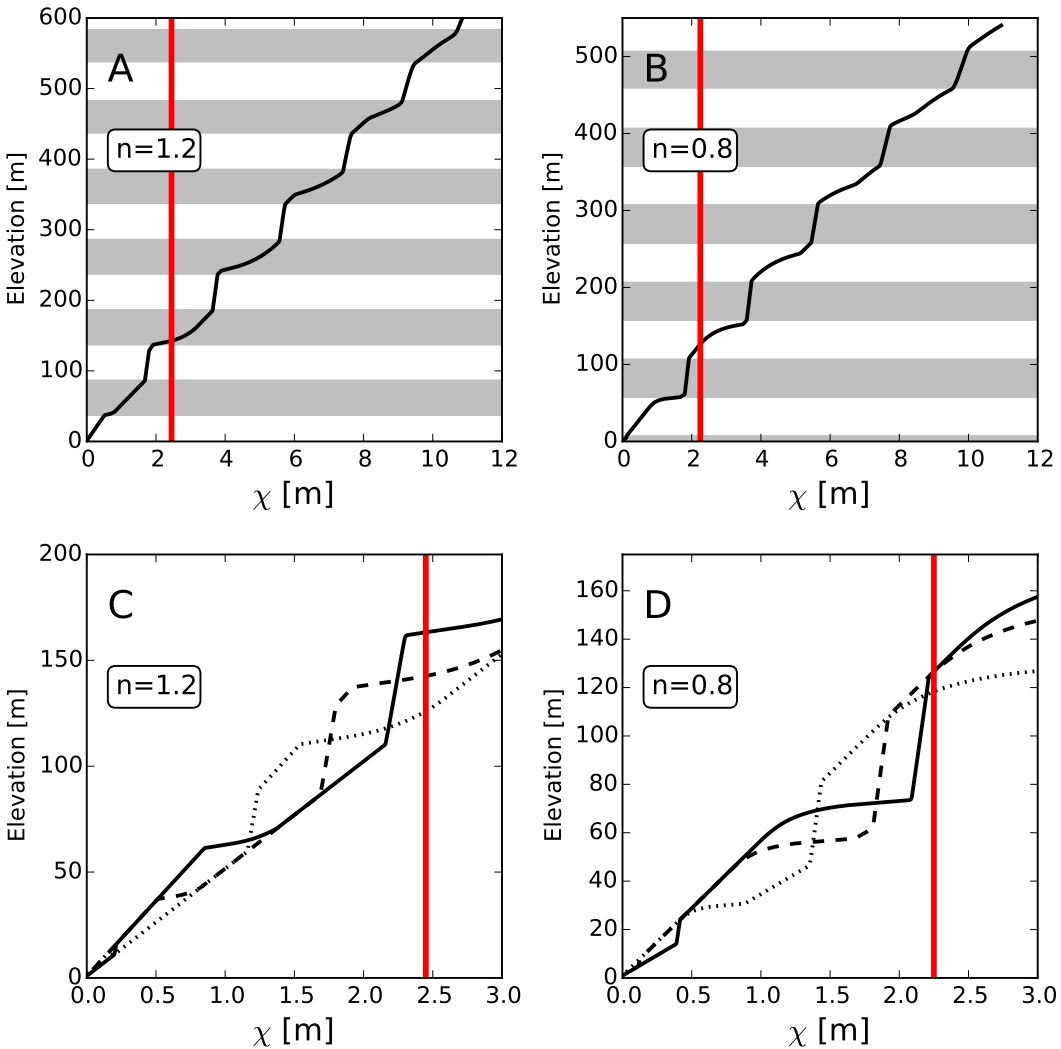

**Figure 9.** Simulations of knickpoint propagation and damping from base level. Entire equilibrium profiles are depicted for cases where $n = 1.2$ (A) and $n = 0.8$ (B). Panels C and D show zoomed figures that depict three separate timesteps (dotted, dashed, and then solid) as fast knickpoints catch up with slow knickpoints at the calculated damping length scale ($\lambda$, thick red line). The interaction of the two knickpoints can be visualized as the reduction in size of a slope patches that are at the topographic equilibrium slope as these patches approach $\lambda$.

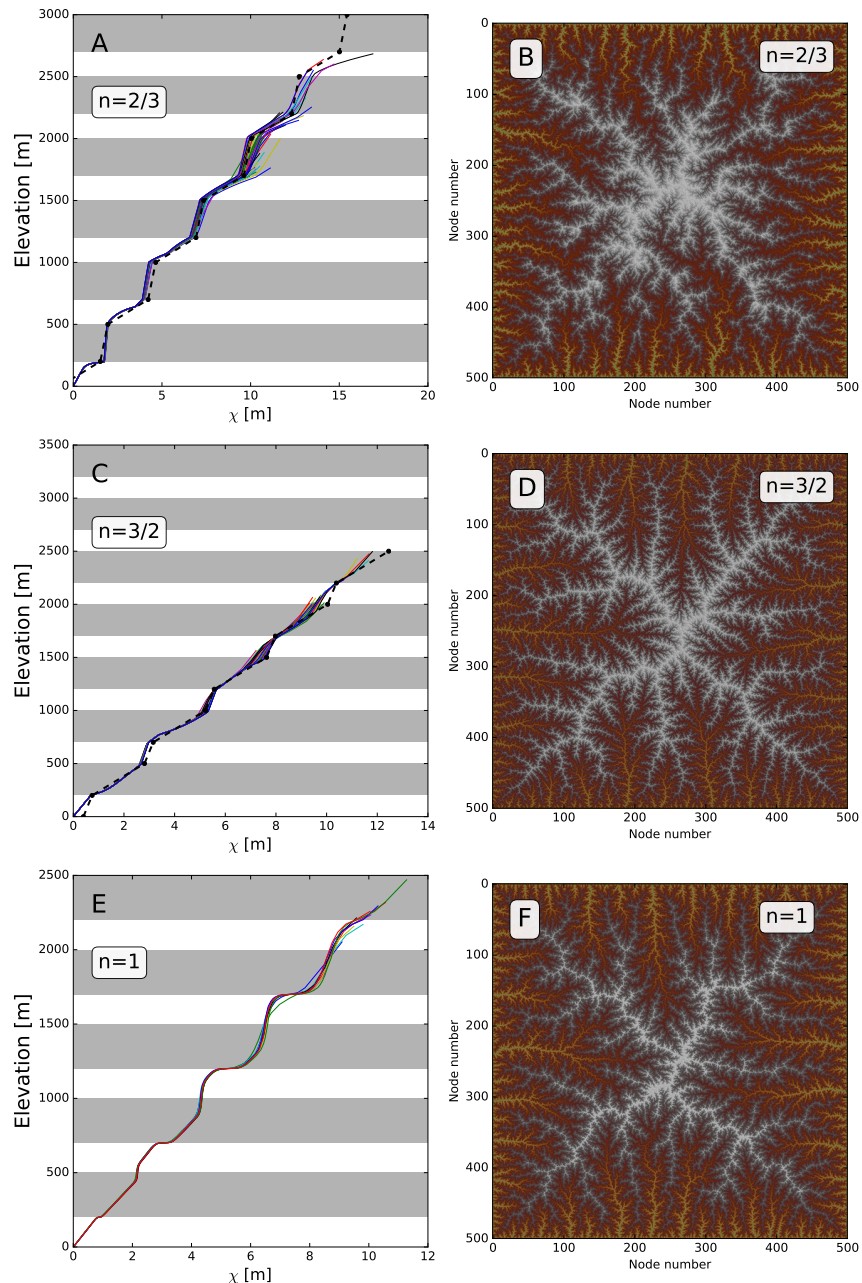

**Figure 10.** Results of the *FastScape* simulations. Lines in the left-hand panels are profiles extracted from the DEMs. Simulations were run at constant uplift with alternating bands of weak and strong rocks. Grey bands indicate the weaker rocks. The individual panels show simulations where $n = 2/3$ (A), $n = 3/2$ (C), and $n = 1$ (E). The dashed lines (A,C) show the equilibrium profile predicted by the theory, with circles depicting predicted crossing points of the contacts. Profiles obtain similar shapes as in the 1D simulations (Figure 2B-D). Panels B, D, and F show DEMs of the landscapes formed in each simulation. Color represents elevation with white being high.

| Simulation | $K_{fw}\,[\mathrm{m}^{1-3m}\,\mathrm{a}^{-1+m}]$ | $K_{fs}\,[\mathrm{m}^{1-3m}\,\mathrm{a}^{-1+m}]$ | $m$ | $P\,[\mathrm{m}\,\mathrm{a}^{-1}]$ | $U\,[\mathrm{m}\,\mathrm{a}^{-1}]$ |
|---|---|---|---|---|---|
| $n=2/3$ | $1.2\cdot10^{-4}$ | $0.5\cdot K_{fw}$ | $1/3$ | $1$ | $2.5\cdot10^{-3}$ |
| $n=1$ | $1.5\cdot10^{-5}$ | $0.83333\cdot K_{fw}$ | $1/2$ | $1$ | $2.5\cdot10^{-3}$ |
| $n=3/2$ | $1\cdot10^{-6}$ | $0.5\cdot K_{fw}$ | $3/4$ | $1$ | $2.5\cdot10^{-3}$ |

**Table 2.** Parameters used in the *FastScape* model runs.

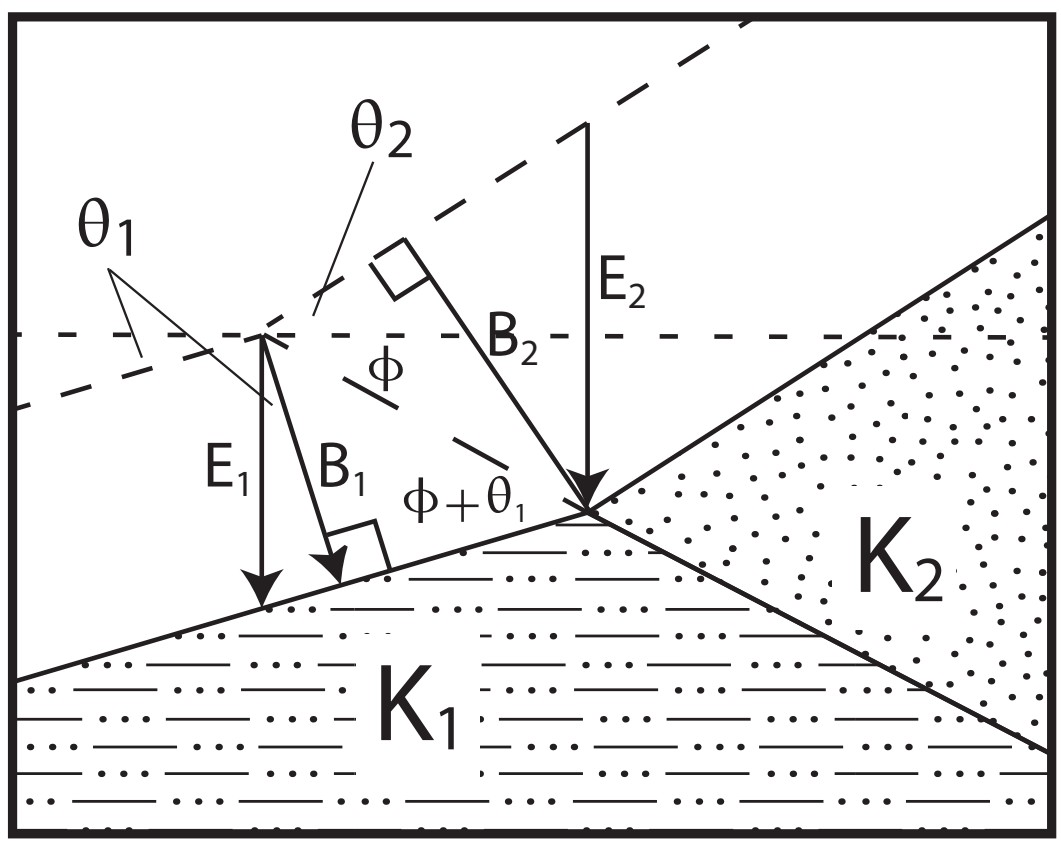

**Figure A1.** Geometric relationships used to derive the equation for continuity of the channel at a contact between two rock types. Note that the slope of the contact plane ($S_c = -\tan\phi$) is defined as positive when the contact dips in the downstream direction.