# Peer review of "Steady state, erosional continuity, and the topography of landscapes developed in layered rocks"

_Earth Surface Dynamics, 2016_

## Referee Comment (RC1) · Anonymous Referee #1 · 11 Sep 2016

This manuscript describes the formulation of a fluvial erosion model based on the principle of continuity, which as described, results in a type of flux steady state. It explores the dynamics of the commonly used stream-power erosion law within this framework, and arrives at a useful new reference case for fluvial analysis in the face of strong contrasts in rock erodibility. This is an important and common geological situation that has until recently received little attention compared to uniform-erodibility cases, perhaps because of its intrinsic complexity. This work contains a thought-provoking analysis that provides a framework for addressing this complexity.

I find the manuscript to be fairly mature and close to ready for publication. I have no major concerns about the formulation or the interpretations, but below point out a few places where further discussion may be warranted. There are a few scattered typos, etc., which I list last.

[Figure]

p2 L28: what does "steady state form of a landscape" mean here? You've just convinced me it doesn't exist in these settings... this is a bit more clear that you mean something like a flux steady state after reading the rest of the paper, but it seems that there is no steady landscape form except in the vertical-contacts case.

p3 L12 "numerical models are more likely to maintain continuity..." this is a good point.

p3 L19 A change in process (e.g. away from stream-power erosion) under steep conditions breaks this relationship, as noted above on L10 or so. This is discussed to some extent but could bear more emphasis. These boundaries are the very places where erosion processes are changing. For example, some of the same authors have published on how blocky debris from strong lithologies locally alters the erosion by streams in these settings. The change to effectively a transport-limited system may necessitate at least a change in the exponents, if not the form, of the erosion law. It is clear from the later discussion that the authors appreciate this; it would be useful at this point perhaps to point out that the formulation in Eq. 3 is effectively a reference case, deviations from which may reflect the process variability present in any particular landscape.

p4 L15 What is considered "subhorizontal" here? How close to horizontal can the contact be before this singularity becomes important? It is rare in nature (but common in LEMs) to have a perfectly uniform, mathematically horizontal dip over a significant distance. I suggest adding an extra set of lines (or two) to Fig. 3 with some dip cases close to horizontal, perhaps $5°$ and $10°$ dip, in addition to the vertical and pure horizontal cases.

p4 L18 "solely a function of erodibility." ... In this framework. I would argue that process variation is critical here. There is certainly field support for a retreat rate that is independent of slope but a function of drainage area in relevant landscapes, a la Crosby & Whipple 2006 (cited) and Berlin & Anderson 2007 JGR (not cited but quite relevant). But another way to view this singularity is that perhaps $n \sim 1$ works well away from contacts in sub-horizontal rocks but the stream power erosion law itself is not a good

model in these situations.

As noted, this is also where numerical inaccuracies may become very important in LEMs. I appreciate the authors pointing out where numerical models may diverge from reality when considering this continuity framework.

p6 L13 "time-averaged incision rate through both rock types..." This needs some clarification. Do you mean vertical incision rate in both rocks is identical to the uplift rate? That doesn't seem quite right. Averaged over what time period?

p6 L17-18 "continuity state is a type of flux steady state" Here this is presented as if it follows from the above analysis, but it was stated on line 13 above that the analysis is based on assuming flux steady state. It reads as being a circular argument, but perhaps the phrasing just needs some clarification.

p7 This analysis of the damping behavior is very interesting with strong field implications.

p7 L27 "two cycles through the rock layers" not clear what this means - what cycles? The perturbation has traversed two sets of contacts?

p7 L30 how does layer thickness affect this result? Presumably it affects the distances across which a profile is developed in each rock type. A common geological scenario is thinner layers of hard rock between thick layers of soft rock. Will thin layers of hard rock slow down knickpoints for less time than thick ones, reducing the damping lengthscale? The analytical expressions and 1D modeling here stick to equal thicknesses of each type. I suspect the general result is the same, but pointing out the effect would be useful, and how to account for it in the framework described on p7.

I see this issue is addressed to some extent in the 2D model setup, but its effect is not then discussed, and the 200 and 300 m alternating thicknesses are similar enough that I wouldn't expect a big impact. What about 100 m of weak rock alternating with 10 m strong-rock interbeds?

p10 L4-5 It's pretty hard to call the reach corresponding to a caprock waterfall a "channel", especially once flow is detached from the face. I think eSurf gives you the space to elaborate a bit more on how processes might commonly change in these settings (see my notes above) and how in general one would incorporate this into the continuity framework (without detailed exploration of such a case).

Minor notes

p2 L17: responce -> response

Fig 7 caption is missing punctuation at the end.

---

## Referee Comment (RC2) · Anonymous Referee #2 · 13 Sep 2016

This manuscript on the influence of horizontally (or close to it) layered rocks and their influence on landscape evolution is very interesting. Some of the results are extremely counter-intuitive, and that always makes for a fun read. The math and modeling seem sound to me, and I'm generally supportive of this paper.

The paper is timely, as another paper on a similar topic recently came out – Forte et al., which is cited here. Forte et al., also discussed that steady state is not reached with horizontal layers. Where this paper falls a bit short, in my opinion, is a lack of much discussion and also some lack in details of the modeling. As for the discussion, I thought they might tie in more with the Forte paper at some point, but that never happened. But in general I did not find the discussion to be very deep. As for the modeling, it was not always clear to me why the models were set-up as they were.

[Figure]

My general comment is just to give a bit more detail, including around the figures, and some suggestions for this are laid out in my line-by-line comments.

Line by line comments:

After reading the abstract I'm still not sure what channel continuity means. Notably, the sentence starting on line 5 made no sense to me, and I think that made me stumble through the rest of the abstract. I went back and read it after reading the manuscript and then it made sense to me. I think it was hard for me to envision what retreat in the direction parallel to a contact meant without the schematics, but after seeing the schematics it seems obvious. I don't have a great suggestion for improving this sentence.

The caption in Figure 2 and main text around it confuse me. In A, is the upper layer steeper, or is it simply that the upper layer is overhanging the lower layer, creating an instability? Similarly, in B, isn't the problem that there was a dam created?

Equation 2: Is this vertical incision rate?

Page 4, first paragraph. I see the math, but this is confusing. A few things. I wonder if it would be helpful to remind people the relationship between $K_w$ and $K_s$? As for equation 5 with $n<1$, the prediction is so counter to my 'gut', that I wonder if some discussion about whether $n<1$ is realistic, or about whether this counter intuitive relationship has been observed, would be useful. The $n=1$ case is also difficult for me to wrap my head around. Maybe more discussion is coming later.

Page 4, line 28: What does it mean that experiments with resolution suggest that the conclusions are not affected by numerics? Does that mean you changed the resolution and ran with different numerical schemes, and got the same answer? Or that your results are not dependent on the resolution for a given implementation of stream power? Please clarify.

Page 5, line 6, 7: Is layer thickness thought to vary with uplift? I don't think so. Why do

you do this?

Page 5, L 14: What do you mean it holds if 'slope is replaced with slope'?

Figure 4 caption: What is meant by the 'steady state profile predicted by the theory'? Just the elev-chi plot for a channel with that erodibility in vertical layers? Or is it the theory that you present in this paper. I'm confused.

Page 7, summary in paragraph on line 25: I got a bit lost. I think a bit more description/hand holding for the reader would help. I recognize that lambda* is a way to show how large chi_s,+ is. But in the description with respect to figure 6, the damping is described in terms of cycles through rock layers. I don't understand what this means, or how to get that from the equations. I must be missing something easy. How does chi_s,+ related to the depth of the rock layers? How do I know from lambda* how many layers the knickpoint has propagated through?

Figure 7 is difficult for me to interpret. I think I can see the knickpoint that is propagating up in elevation, but I can't really make out the knickpoint that it is 'catching'. Can you tell us how you determined that there was a knickpoint at the red line that was caught? If I look at the dashed line (intermediate time) in C, it does not look like there is any significant change in the chi-elev relationship at the red line, but I think that there is supposed to be a knickpoint there, right? Or at least one close to it that will soon catch up? I only see one knickpoint downstream from there, but maybe I am interpreting incorrectly? Actually, after watching the movies, I may understand this. But I still think it is worthwhile to point out to readers exactly what you are calling knickpoints.

Fastscape runs: It is a bit unsatisfying that the n=2/3, 3/2 runs have channels that extend through 4+ layers of each rock type, but the n=1 run only just barely taps three week layers. I know this is a lot to ask, but it'd be more satisfying to see more of the n=1 profiles, i.e. just make the K values in this run smaller. I'm not adamant about this, as the 2D runs appear to be very similar to the 1D runs.

In the beginning of Section 4, the authors mention that they include hillslope processes. It seems like this needs a bit more description. How are the different rock types treated with the hillslope model? How do they model hillslopes?

Discussion and Conclusions: I liked that the authors brought in a real world example. However, this example confused me. I may be wrong, but my impression of Niagara Falls and the Niagara river is that the soft rocks underneath the hard caprock are indeed basically vertical at the waterfalls. But if you move any length downstream the channel is not so steep anymore. I might be wrong as I haven't studied the Niagara River, just visited it. But does the whole length of profile have the 'inverted' relationship (steeper in weak rocks) suggested by Figure 4D, or is it just 'inverted' around the waterfall? This may seem a picky point, but I would guess, as the authors brought up elsewhere, that the processes going on right at the knickpoint are not adequately modeled by stream power. So in some ways this comparison feels a bit odd to me.

I felt as though the discussion could be expanded a bit. The parameter n turns out to be extremely important in this study. Any thoughts beyond Niagara Falls on how your contribution plays in to the n debate? Have many studies suggested that n<1? Are there any other landscapes to call upon to illustrate the modeled behavior besides Niagara Falls?

I also generally prefer a separate conclusions section. I think it is better for authors because often times the only sections of the paper that get read are the abstract and conclusions. But this is stylistic.

---

## Referee Comment (RC3) · K. Whipple (Referee) · 14 Sep 2016

This manuscript is well written and entails an important step forward in understanding the influence of rock strength variations in landscape evolution. The novel focus is on the influence of the slope exponent (n) in the stream power river incision model on landscape evolution in areas where sub-horizontal layered rocks with varying rock strength are exposed – extending beyond a recent treatment from my group (Forte et al., 2016, Earth Surface Processes and Landforms) that considered only the n = 1 case. It is remarkable that the venerable stream power model still holds surprises! Though of course it is always important to consider the degree to which processes and effects not encapsulated in the stream power model will alter the behavior of natural landscapes.

There is much value in the analysis and discussion presented. Reading and carefully reviewing this paper has notably advanced my own understanding of how landscapes described by the stream power model will evolve in the presence of layered rocks as a function of the relative strength between stronger and weaker layers, the relative thickness of strong and weak layers, and the dip of the contacts (only simple planar dip panels considered thus far) in cases with n<1 or n>1.

As part of the process of reviewing this paper I re-derived most of the key relationships and updated an existing 1d finite-difference solver to handle a series of dipping layers with variable erodibility (K in the stream power model) and variable thickness so I could test both the author's initially counter-intuitive results (such as the formation of cliffs in the weak units, not the strong units, if n < 1) and my own derivations. I find complete agreement with Figures 3, 4, 5, and 8. Similarly, though I would word some aspects differently (reflecting differences in my derivations described below), I agree with the points made in the discussion and conclusions. Thus I agree with all the findings in a qualitative sense. Likewise I see no problems with the numerical simulation results – both in 1d and 2d using FastScape.

However, I do not agree with some of the derivations and prefer a different approach to solving the problems discussed and explaining the interesting results of the 1d profile evolution models. As the only way I felt I could evaluate the derivations was to re-do them following my own intuition for how to pose the problem, I present alternative solutions below. Rather than working the derivations here, I outline the logic the present the solution. Hopefully this will prove an effective and constructive approach. The alternate derivation given below results in an identical solution for horizontal bedding (Eqn 5), which is good, but suggests differing sensitivities to the dip of contacts and the relative thicknesses of strong and weak units.

First, I don't much like the conceptual model in Figures 1 and 2. Most important, a problem only arises in the strong-over-weak case: overhangs cannot be sustained, as illustrated in Figure 2a. Conversely, as illustrated by Forte et al. (2016) and commonly

seen in nature, weak rocks can readily be stripped off the top of strong rocks, leaving a tapering wedge of weak rock in the case of an upstream-dipping contact like that shown in Figure 2b. I also don't like the use of the word "continuity" for this, since in much of the geomorphic and fluid flow literature "continuity" means conservation of mass, though I appreciate that you are imposing a continuous profile with no overhangs.

I find it most useful to think about this problem in terms of the controls on the kinematic wave speed that characterizes the evolution of river profiles governed by the stream power incision model (Rosenbloom and Anderson, 1994): $C_e = K A^m S^{(n-1)}$. Key elements are (1) all else equal the kinematic wave speed is higher in weak rocks than strong, and (2) wave speed decreases with Slope for n<1, is independent of Slope for n=1, and increases with Slope for n>1. The surprising results in this paper all stem from the curious effect that wave speed decreases with Slope for n<1.

From study of the evolution of 1d river profiles cutting through layered rocks for cases n<1, n>1, and n=1 revealed in numerical simulations (as in Figures 4 and 5), I suggest below a set of fundamental controls on the development of profile shape (cliffs formed in the weak rock (n<1), the strong rock (n>1), or through each strong-over-weak couplet (n=1)), and the retreat rate of the slope-break knickpoint at the strong-over-weak contact.

The authors come close to stating what I believe is happening in the case of horizontal contacts: (1) fundamentally cliffs are forming because all-else held equal the kinematic wave speed of profile segments within the weak unit exceeds that of segments within the strong unit, so there is a tendency to undermine, or to form consuming knickpoints at strong-over-weak contacts, but as described by the authors and illustrated in the numerical simulations, the river profile will evolve toward an equilibrium where the upstream migration rate of the strong unit matches that of the weak unit at the contact; (2) for n<1 wave speed decreases with increasing slope, so in response to the tendency to undermine, the profile steepens in the weak unit until the wave speed of the weak unit at the contact has slowed to equal the wave speed of the strong unit at the contact

(the strong unit maintains an equilibrium slope at the contact and the knickpoint at the contact migrates at the rate set by the wave speed of the strong unit on an equilibrium slope – this best describes the basal strong-over-weak contact, see below); (3) for n=1 wave speed is independent of slope, so the river profile has no way to respond, and a vertical cliff forms (50% in weak, 50% in strong unit) with retreat rate = wave speed of the weak unit – the cliff grows in height until the full strong-over-weak doublet is incorporated and the retreat rate is set by the wave speed of the weak unit; (4) for n>1 wave speed increases with increasing slope, so in response to the tendency to undermine, the profile steepens in the strong unit until the wave speed of the strong unit at the contact has increased to equal the wave speed of the weak unit at the contact (the weak unit maintains an equilibrium slope at the contact and the knickpoint at the contact migrates at the rate set by the weak unit on an equilibrium slope).

Once this realization is made, it is easy to write equations for the wave speed in each unit at the contact, set them equal, and solve for the ratio of the slope of the weak unit ($S_w$) to the slope of the strong unit ($S_s$). For horizontal beds, Equation 5 in the paper is recovered. So the derivation given is exact in the limit of horizontal contacts (also satisfies expectation for vertical contacts). However, in my analysis the derivation for the case of non-horizontal contacts (Eqn 2) appears to be incorrect. First, the solution only applies for strong-over-weak scenarios, so $E_1$, $S_1$ (downstream) could only be the weak unit. Second, if the derivation described above for horizontal contacts is generalized to account for planar dipping beds, a different solution is found.

In the case of dipping beds, the migration rate of the knickpoint at the strong-over-weak contact is not set only by the kinematic wave speed ($C_e = E/S = K A^m S^{(n-1)}$) as it is for horizontal contacts, but must account for the slope of the contact ($S_c$). For example, if the contact were exactly parallel to the river bed, then the migration rate would approach infinity over that reach. Geometrically one can readily show that the local knickpoint migration velocity ($C_{e\_kp}$) will be: $C_{e\_kp} = E/(S – S_c)$ (which correctly reduces to the kinematic wave speed for $S_c = 0$).

(Note here that the caption to Fig. A1 indicates that Sc in the paper is defined positive for an upstream dip, while channel slope S is positive downstream. I worry that this could prove very confusing. Here I instead define Sc as positive for downstream dip).

As noted above, the migration rate of the slope-break knickpoint at the contact is set by the equilibrium wave speed within the strong unit for n<1 (at least for the basal strong-over-weak contact), and within the weak unit for n>1 – the problem then is how the dip of the contact amplifies or reduces knickpoint velocity relative to the kinematic wave speed. Solving for the equivalent of Eqn 5 in the presence of dipping beds, I find (derivations available on request):

Sw/Ss = (Kw/Ks)^(1/(1-n)) * (1 − Sc/Ss)^(1/(1-n)) ; for n<1 Sw/Ss = (Kw/Ks)^(1/(1-n)) * (1 − Sc/Sw)^(1/(n-1)) ; for n>1

Note that Sc/Ss appears in the n<1 case because the retreat rate is set by the wave speed in the strong unit, and Sc/Sw appears in the n>1 case because the retreat rate is set by the wave speed in the weak unit.

These solutions, however, only obtain over a range of Sc/Ss or Sc/Sw – basically restricted to sub-horizontal conditions – as outlined below. In addition, as mentioned above, the n<1 solution applies best to the basal strong-over-weak contact: the over-steepening of the weak unit is damped up-section because the slope-break knickpoints at the strong-over-weak contacts act as a local baselevel, reducing local incision rate within the overlying strong unit (as happens in weak-over-strong contacts with n=1). This causes a decrease in the slope within the strong unit, which increases the kinematic wave speed and thus decreases the degree of over-steepening of the weak unit. This complicating phenomenon is restricted to the n<1 case.

I have tested these revised equations, and the limits on their applicability outlined below, against numerical simulations with satisfying results.

For n<1 and downstream-dipping beds (Sc is positive), the solution only applies for

Interactive
comment

[Figure]

Sc/Ss < 1 – (Ks/Kw)^(1/n): for larger (more positive) downstream dips, an equilibrium profile results (Ss and Sw have equilibrium values = to the vertical contact case even though knickpoints are slowly migrating upstream over time). For n<1 and upstream-dipping beds (Sc is negative), preliminary comparison with numerical simulations indicates the solution is only valid for abs(Sc/Ss)<~1. For steeper upstream dips, the profile transitions toward an equilibrium form (I have not studied this in detail).

For n>1 and downstream-dipping beds (Sc is positive), the solution only applies for Sc < Sw. At Sc = Sw, knickpoint velocity is infinite. For Sc > Sw, the strong-over-weak contact propagates downstream, invalidating the analysis. For n>1 and upstream-dipping beds (Sc is negative), the solution only applies for Sc/Sw > 1 – (Kw/Ks)^(1/n): for larger (more negative) upstream dips, an equilibrium profile results (Ss and Sw have equilibrium values = to the vertical contact case even though knickpoints are slowly migrating upstream over time).

These solutions can be re-cast into the form of Eqn 2 (note I have inverted the relation here):

E2/E1 = Es/Ew = (Ss – Sc)/Sw ; for n<1  E2/E1 = Es/Ew = Ss/(Sw – Sc) ; for n>1

Thus Eqn 2 should have two forms, one for n < 1 and one for n > 1. (remember that Sc is defined here as positive downstream).

Section 3.2. I did not attempt to reproduce or critically evaluate Equation 8, but found no dependence of erosion rate patterns on H1/H2 in my numerical simulations. For horizontal beds, Equation 5 is exactly satisfied for a very wide range of H1/H2. I did not investigate whether a greater sensitivity to layer thickness emerges with dipping contacts.

Section 3.3. I don't see the profile as being "perturbed" at baselevel because, as the authors note on page 6, line 27, new river segments formed at baselevel always begin in equilibrium (E = U, and equilibrium slopes). The perturbations develop above as

the differential wave speeds near contacts begin to manifest in deviations from equilibrium slopes and erosion rates. Thus I'm not enthused about the "damping length scale" terminology. However, the result appears robust – differential wave speeds are rapidly accommodated at the first strong-over-weak contact, with knickpoints at contacts quickly converging on a migration velocity set by the equilibrium wave speed of either the weak unit ($n \geq 1$) or the strong unit ($n < 1$).

That said, I am confused by Eqn 10. First, there appears to be a typo in Eqn (10): as derived, the last term should be $A_o^{(m/n)}$ not $A^{(m/n)}$. Further, $C_e$ = kinematic celerity = horizontal migration rate of river "patch" (patch as used by Royden and Perron, 2012) = $K\,A^m\,S^{(n-1)}$. For a steady-state river patch, $S = (U/K)^{(1/n)}\,A^{(-m/n)}$. Combining these, Whipple and Tucker (1999) showed that the horizontal migration rate of a steady-state river patch is $C_e = U^{((n-1)/n)}\,K^{(1/n)}\,A^{(m/n)}$ – this is the relation given for Eqn 10, so the equation appears to be correct, but the derivation (and the apparent typo) implies it is incorrect.

Finally, although widely appreciated, it seems worth stating that readers should beware the difference between the mathematics of the stream power model (SPM), insightful though they can be, and the physical reality of nature. Many processes are not represented in the SPM and therefore predictions may fail. Despite this, I am very supportive of publishing papers like this that explore model predictions because this allows one to: (1) generate testable hypotheses, constrain parameters, or recognize where models fail and why; (2) use any failures to improve the model; and (3) know what will happen in landscape evolution simulations based on the SPM under different conditions.

I have a few additional comments listed below with reference to page and line number.

1. Title: I suggest revising title to remove "continuity" as this will mean "conservation of mass" to many. Also I suggest emphasizing your key finding about the dependence on n, if you can find an effective wording.

2. Page 1, Line 21-22: This is not true. Many studies of bedrock channel morphology

are expressly seeking information about the history of climate, tectonics, or drainage divide migration recorded in non-steady state profiles (as you note on page 2, line 4).

3. Page 2, line 9: better to not call the stream power model (SPM) a "law".

4. Page 2, line 11-12: the SPM is widely used in modeling studies, but is not required as a basis of profile analysis – channel steepness and concavity can be measured and interpreted in terms of relative uplift rate, climate, or rock strengths independent of the river incision rule.

5. Page 3, line 3-4: as you show in your analysis, this is not true for n<1.

6. Page 3, line 5-6: I disagree. Where a weak layer overlies a strong layer, there is no constraint on the relative stream segment migration speed – the weak layer can be stripped off, leaving a bench on the underlying strong layer or a tapering wedge of the weak layer. Such forms are very common in nature.

7. Page 9, line 9: This sentence is confusing since channel segments formed at base-level are always initially at equilibrium (E=U, steady state form) in systems described by the SPM.

8. Page 9, line 16: "channel steepness" here would be better written as "channel slope" or "channel gradient", since "steepness" commonly refers to the steepness index.

---

## Author Comment (AC1) · 16 Sep 2016

We thank the reviewer for their careful criticism of the text and inciteful comments that have helped us to clarify and improve the manuscript. We give detailed responses to specific comments below.

**p2 L28: what does "steady state form of a landscape" mean here? You've just convinced me it doesn't exist in these settings...this is a bit more clear that you mean something like a flux steady state after reading the rest of the paper, but it seems that there is no steady landscape form except in the vertical-contacts case.**

We changed the wording in this sentence, and added an additional sentence to clarify that we are talking about a flux steady state rather than topographic steady state.

[Figure]

**p3 L19 A change in process (e.g. away from stream-power erosion) under steep conditions breaks this relationship, as noted above on L10 or so. This is discussed to some extent but could bear more emphasis. These boundaries are the very places where erosion processes are changing. For example, some of the same authors have published on how blocky debris from strong lithologies locally alters the erosion by streams in these settings. The change to effectively a transport-limited system may necessitate at least a change in the exponents, if not the form, of the erosion law. It is clear from the later discussion that the authors appreciate this; it would be useful at this point perhaps to point out that the formulation in Eq. 3 is effectively a reference case, deviations from which may reflect the process variability present in any particular landscape.**

We agree with the reviewer on this point and have added a couple of sentences to make this assumption explicit.

**p4 L15 What is considered "subhorizontal" here? How close to horizontal can the contact be before this singularity becomes important? It is rare in nature (but common in LEMs) to have a perfectly uniform, mathematically horizontal dip over a significant distance. I suggest adding an extra set of lines (or two) to Fig. 3 with some dip cases close to horizontal, perhaps 5 and 10 dip, in addition to the vertical and pure horizontal cases.**

Subhorizontal is defined on Lines 1-2 of page 4. It is whenever rock dip is small compared to channel slope. Therefore, the cases shown actually span a wide range of possible contact and channel slopes. There is not a simple way that we can think of to show specific other choices of dip angle. We have modified the main text and figure caption to make it clearer that these two limits are not explicitly a function of rock dip, but rather a comparison between rock dip and channel slope.

**p4 L18 "solely a function of erodibility." In this framework. I would argue that process variation is critical here. There is certainly field support for a retreat**

**rate that is independent of slope but a function of drainage area in relevant land-scapes, a la Crosby Whipple 2006 (cited) and Berlin Anderson 2007 JGR (not cited but quite relevant). But another way to view this singularity is that perhaps n=1 works well away from contacts in sub-horizontal rocks but the stream power erosion law itself is not a good model in these situations. As noted, this is also where numerical inaccuracies may become very important in LEMs. I appreciate the authors pointing out where numerical models may diverge from reality when considering this continuity framework.**

We agree. For n = 1 the horizontal retreat rate is a function of erodibility AND drainage area and independent of slope. This is a direct consequence of stream power ero-sion law. (In chi space, for n = 1 the horizontal retreat rate is a function of erodibility and independent of slope AND recharge area.) We have corrected the text to include drainage area as a factor influencing retreat rate. We also agree that the singularity precludes validity of the stream power erosion law in these situations because it causes the predicted slopes to not be small enough. We have also added a couple of lines of discussion concerning the cited field work and implications for the n=1 case.

**p6 L13 "time-averaged incision rate through both rock types: : :" This needs some clarification. Do you mean vertical incision rate in both rocks is identical to the uplift rate? That doesn't seem quite right. Averaged over what time period? p6 L17-18 "continuity state is a type of flux steady state" Here this is presented as if it follows from the above analysis, but it was stated on line 13 above that the analysis is based on assuming flux steady state. It reads as being a circular argument, but perhaps the phrasing just needs some clarification.**

This section was not very clear and did appear circular. We have edited it to make it clearer. From the results of the simulations, and specifically the fact that the landscape is periodic in chi space, you can argue that the system must be in a flux steady state. Using this conclusion, we can derive full profiles. Finally, that the whole story holds together is further confirmed by the fact that we can match the simulated profiles using
the equation derived from flux steady state.

**p7 L27 "two cycles through the rock layers" not clear what this means - what cycles? The perturbation has traversed two sets of contacts?**

Not exactly - it means the knickpoint caused by the perturbation has travelled so far upstream that two sets of contacts now separate it from the downstream end of the channel that is being perturbed. The number of contacts it traversed on its way (if any) depends on the ratio between horizontal retreat rates and knickpoint celerity. As a side note, knickpoints pass from one lithology to another unobstructed. They get damped through formation of stretch zones as in Royden Perron 2013 and through interfering with one another. We have edited the text to try to clarify this point.

**p7 L30 how does layer thickness affect this result? Presumably it affects the distances across which a profile is developed in each rock type. A common geological scenario is thinner layers of hard rock between thick layers of soft rock. Will thin layers of hard rock slow down knickpoints for less time than thick ones, reducing the damping lengthscale? The analytical expressions and 1D modeling here stick to equal thicknesses of each type. I suspect the general result is the same, but pointing out the effect would be useful, and how to account for it in the framework described on p7. I see this issue is addressed to some extent in the 2D model setup, but its effect is not then discussed, and the 200 and 300 m alternating thicknesses are similar enough that I wouldn't expect a big impact. What about 100 m of weak rock alternating with 10 m strong-rock interbeds?**

Only the thickness of the stronger layer influences this length scale. This results because the problem is asymmetric with respect to the two rocks. The strong rock knickpoints are always slower. The time for the weak knickpoint to catch up depends on only three things: 1) how big of a head start the strong knickpoint has, 2) the velocity of the weak knickpoint, 3) the velocity of the strong knickpoint. The velocities of the two knickpoints are independent of layer thickness. The head start of the strong knickpoint

is only dependent on the thickness of the strong rock. Therefore, the thinner the strong rock layer, the quicker the knickpoints should decay.

We agree that it would be interesting to simulate some cases with thin, hard layers. We will begin simulating such cases for possible inclusion within the manuscript.

**p10 L4-5 It's pretty hard to call the reach corresponding to a caprock waterfall a "channel", especially once flow is detached from the face. I think eSurf gives you the space to elaborate a bit more on how processes might commonly change in these settings (see my notes above) and how in general one would incorporate this into the continuity framework (without detailed exploration of such a case).**

We agree that processes dramatically change in this setting. Our speculation in the manuscript is that stream power erosion, specifically in subhorizontal rocks with n<1 is one possible mechanism to drive the system toward the caprock waterfall state. Once the system reaches this state, stream power erosion has certainly broken down. We are considering how we might further elaborate on these ideas and will include more detail in our final reviewer response.

**Minor notes p2 L17: responce -> response Fig 7 caption is missing punctuation at the end.**

These typos were corrected.

---

## Short Comment (SC1) · 19 Sep 2016

We thank Whipple for his careful examination of our work, and the large amount of time that he has obviously put in to try to understand it (even running his own simulations). Undoubtedly, these comments will help to clarify and correct our manuscript. Before we post a more general reply, we want to make sure that we correctly understand the explanation that Whipple puts forward. This is our understanding of his explanation:

1) In a layered rock scenario, either the strong (if $n<1$) or weak (if $n>1$) layers control the horizontal retreat rates of a channel at the contact.

2) The kinematic wave speed of the non-controlling rock layer adjusts to match the kinematic wave speed of the controlling rock layer.

[Figure]

3) The controlling rock layer maintains the same slope and kinematic wave speed that it would have under equilibrium conditions if it were the only rock layer (this is the point that we were least certain of).

Do we understand correctly? If so, these ideas should be fairly easy to test from our model outputs.

---

## Referee Comment (RC4) · K. Whipple (Referee) · 21 Sep 2016

Yes, you understand my argument correctly.

In my view, what you are concerned about in point 3 is valid - the solution I provided is an approximation. From running numerical simulations I could perceive that channel profiles with the "controlling layer" near the contact maintained equilibrium or near equilibrium slopes. This observation provided the insight for an analytical solution. Where this solution could fail is if the influence of lower non-controlling layers in the sequence causes the controlling layer to deviate from an equilibrium slope and kinematic wave speed. Indeed I do see this happening in the n<1 case, but not in the n>1 case in my simulations. This is why I say the n<1 case applies best to the basal strong-over-weak contact. In layers above I find the channel profiles in the strong layers have somewhat

reduced slopes and thus faster wave speeds, which results in lower slopes on the over-steepened weak layers as well – a progressive decline with each strong/weak pair. A damping of the signal related to the strong over weak contact if you will.

I did not pursue this aspect further. One might imagine a similar complication for n>1 but I didnt detect one in my simulations.

Either way, the approximate analytical solution, I believe, provides useful guidelines and a closer approximation to system evolution in the presence of dipping contacts than provided by your original derivation. Naturally either solution only pertains to the stream power river incision model and does not account for some of the processes likely to influence the evolution of river profiles in the vicinity of a strong-over-weak contact.

---

## Author Comment (AC2) · 22 Sep 2016

This manuscript on the influence of horizontally (or close to it) layered rocks and their influence on landscape evolution is very interesting. Some of the results are extremely counter-intuitive, and that always makes for a fun read. The math and modeling seem sound to me, and I'm generally supportive of this paper. The paper is timely, as another paper on a similar topic recently came out – Forte et al., which is cited here. Forte et al., also discussed that steady state is not reached with horizontal layers. Where this paper falls a bit short, in my opinion, is a lack of much discussion and also some lack in details of the modeling. As for the discussion, I thought they might tie in more with the Forte paper at some point, but that never happened. But in general I did not find the discussion to be very deep. As for the modeling, it was not always clear to me why the models

[Figure]

**were set-up as they were. My general comment is just to give a bit more detail, including around the figures, and some suggestions for this are laid out in my line-by-line comments.**

We thank the reviewer for their careful reading of the manuscript, and pointing out a number of items that were unclear or deserved further elaboration. Detailed responses to comments are given below.

**Line by line comments: After reading the abstract I'm still not sure what channel continuity means. Notably, the sentence starting on line 5 made no sense to me, and I think that made me stumble through the rest of the abstract. I went back and read it after reading the manuscript and then it made sense to me. I think it was hard for me to envision what retreat in the direction parallel to a contact meant without the schematics, but after seeing the schematics it seems obvious. I don't have a great suggestion for improving this sentence.**

We agree that it is difficult to understand without a figure, and are not exactly sure how to make it clearer. In general, we are also considering whether there is a better word than continuity (see Reviewer 3 comments) or an easier way to explain the concept. Any changes regarding this point will be detailed in the final reviewer reply.

**The caption in Figure 2 and main text around it confuse me. In A, is the upper layer steeper, or is it simply that the upper layer is overhanging the lower layer, creating an instability? Similarly, in B, isn't the problem that there was a dam created? Equation 2: Is this vertical incision rate?**

We have attemped to make this clearer. In case A, the upper layer can become steeper or create an overhang, it depends on the dip of the contact. In case B, the lower does always create a kind of dam (where presumably sediment could accumulate). We have adjusted the text accordingly.

**Page 4, first paragraph. I see the math, but this is confusing. A few things. I**

**wonder if it would be helpful to remind people the relationship between $K_w$ and $K_s$? As for equation 5 with n<1, the prediction is so counter to my 'gut', that I wonder if some discussion about whether n<1 is realistic, or about whether this counter intuitive relationship has been observed, would be useful. The n=1 case is also difficult for me to wrap my head around. Maybe more discussion is coming later.**

It is definitely counter to the common intuition based on prior work. One of the main points of this manuscript is that the assumption behind that prior work actually can break down in subhorizontal rocks. We try to explain this in lines 6-8 of this page. Basically, it results because horizontal retreat rate (or knickpoint celerity) is lower for steeper channels in the case where n<1. This is because, for the same rate of vertical erosion, horizontal retreat rates are less for steeper slopes. In cases where n<1, the increased erosion from the channel becoming steeper isn't sufficient to offset the slope effect. At n=1, these two effects are totally balanced, so slope has no effect on horizontal retreat rate. We have expanded this section to try to make it a bit clearer. We do also discuss later cases where n<1 might be reasonable.

**Page 4, line 28: What does it mean that experiments with resolution suggest that the conclusions are not affected by numerics? Does that mean you changed the resolution and ran with different numerical schemes, and got the same answer? Or that your results are not dependent on the resolution for a given implementation of stream power? Please clarify.**

Our original statement was a bit too vague. We have edited this to clarify that we ran some higher resolution simulations for some cases that produced the same result.

**Page 5, line 6, 7: Is layer thickness thought to vary with uplift? I don't think so. Why do you do this?**

We are specifically examining cases where there are many different rock layers, such that the influence of base level perturbations dies out and we can see the continuity

equilibrium form. This was just a practical way of generating a similar number of contacts in both the high and low uplift cases (albeit both with parameter ranges that are within the range of natural streams).

**Page 5, L 14: What do you mean it holds if 'slope is replaced with slope'?**

Slope is replaced with "slope in $\chi$-elevation space." We agree that the repeated word obscures the meaning a bit. We have replaced the second "slope" with "gradient" to remove this repetition.

**Figure 4 caption: What is meant by the 'steady state profile predicted by the theory'? Just the elev-chi plot for a channel with that erodibility in vertical layers? Or is it the theory that you present in this paper. I'm confused.**

We mean the theory presented in this paper. We have edited the caption to say "profile predicted by continuity steady state."

**Page 7, summary in paragraph on line 25: I got a bit lost. I think a bit more description/hand holding for the reader would help. I recognize that lambda\* is a way to show how large $\chi_{s,+}$ is. But in the description with respect to figure 6, the damping is described in terms of cycles through rock layers. I don't understand what this means, or how to get that from the equations. I must be missing something easy. How does $\chi_{s,+}$ related to the depth of the rock layers? How do I know from lambda\* how many layers the knickpoint has propagated through?**

$\chi_{s,0}$ is the $\chi$ length of the strong layer reach near base level at the moment that the weak layer becomes exposed at base level. Consequently, this distance is less than the profile distance spanned by a pair of weak and strong rocks, but is also on the same order of magnitude. The dimensionless damping length scale, $\lambda^* = \lambda/\chi_{s,0}$, therefore provides a rough (conservative) estimate of the number of strong/weak pairs that the knickpoint will pass before significant damping. We have expanded this text to clarify this point.

**Figure 7 is difficult for me to interpret. I think I can see the knickpoint that is propagating up in elevation, but I can't really make out the knickpoint that it is 'catching'. Can you tell us how you determined that there was a knickpoint at the red line that was caught? If I look at the dashed line (intermediate time) in C, it does not look like there is any significant change in the chi-elev relationship at the red line, but I think that there is supposed to be a knickpoint there, right? Or at least one close to it that will soon catch up? I only see one knickpoint downstream from there, but maybe I am interpreting incorrectly? Actually, after watching the movies, I may understand this. But I still think it is worthwhile to point out to readers exactly what you are calling knickpoints.**

Figure 7 was the best way we could think of to show this statically, though it is much clearer in the animations, as we state in the text. We think that the point of confusion here is that the knickpoints we are talking about are just sudden changes in slope, and can correspond to increases or decreases in slope (depending on $n$). We have expanded this explanation in the text.

**Fastscape runs: It is a bit unsatisfying that the n=2/3, 3/2 runs have channels that extend through 4+ layers of each rock type, but the n=1 run only just barely taps three week layers. I know this is a lot to ask, but it'd be more satisfying to see more of the n=1 profiles, i.e. just make the K values in this run smaller. I'm not adamant about this, as the 2D runs appear to be very similar to the 1D runs.**

We will try some cases with smaller K to see if we can replace this figure with one that is more similar to the cases where n is not 1.

**In the beginning of Section 4, the authors mention that they include hillslope processes. It seems like this needs a bit more description. How are the different rock types treated with the hillslope model? How do they model hillslopes?**

We are using the standard hillslope diffusion approach employed in Fastscape, which does not have the capability to adjust the diffusion coefficient with rock type. We now

clarify this in the text.

**Discussion and Conclusions: I liked that the authors brought in a real world example. However, this example confused me. I may be wrong, but my impression of Niagara Falls and the Niagara river is that the soft rocks underneath the hard caprock are indeed basically vertical at the waterfalls. But if you move any length downstream the channel is not so steep anymore. I might be wrong as I haven't studied the Niagara River, just visited it. But does the whole length of profile have the 'inverted' relationship (steeper in weak rocks) suggested by Figure 4D, or is it just 'inverted' around the waterfall? This may seem a picky point, but I would guess, as the authors brought up elsewhere, that the processes going on right at the knickpoint are not adequately modeled by stream power. So in some ways this comparison feels a bit odd to me. I felt as though the discussion could be expanded a bit.**

If the stream power erosion law continued to hold as the channel steepened, then in theory one would expect the entire channel to remain steep in the weak rocks (for n<1)). However, the stream power erosion law breaks down for such steep channels as erosion processes take over that are not well-described by the stream power law. Therefore, we are only speculating that continuity can push the system toward this state (by first making the channel steep in the weak rocks). From a more standard assumption of topographic equilibrium, one would never expect steepening in weak rocks, so it is not clear how you would approach such a state to begin with. There are potentially other explanations, such as non-locality in erosion processes near the contact, that cannot be entirely ruled out. We have expanded this discussion slightly and tried to make it clearer that it is speculative. In the specific case of Niagara Falls, which is one of the most famous of many possible examples, the flattening below the waterfall in part occurs because of the nearby base level imposed by Lake Ontario.

**The parameter n turns out to be extremely important in this study. Any thoughts beyond Niagara Falls on how your contribution plays in to the n debate? Have**

**many studies suggested that n<1? Are there any other landscapes to call upon to illustrate the modeled behavior besides Niagara Falls? I also generally prefer a separate conclusions section. I think it is better for authors because often times the only sections of the paper that get read are the abstract and conclusions. But this is stylistic.**

We do not attempt to constrain what realistic values of n should be. There are theoretical arguments that some incision processes will produce n<1 (e.g. Whipple et al. [2000] cited in this work, or Covington et al. [2015], GRL). While there are likely good field sites where a natural experiment could be used to test the ideas developed here, we think that finding and studying such a site is beyond the scope of this manuscript. Our main goal here is to solidify our theoretical understanding of the equilibrium behavior of the stream power erosion law in layered rocks.

We have expanded the discussion and written a separate conclusions section. The expanded discussion focuses on clarifying our ideas concerning caprock waterfalls and on discussing the relationship between our work and that of Forte et al. (2016) more extensively. The lack of a detailed discussion of the Forte et al. paper was an oversight, which partly resulted because we had an initial draft of our manuscript before the Forte et al. paper was published. Citations and brief comments were added after the fact. We now carefully explain the relationship between our work and that of Forte et al.

---

## Author Comment (AC3) · 24 Oct 2016

We thank the reviewers for their careful criticism of the text and inciteful comments that have helped us to clarify and improve the manuscript. We think that these comments have helped us to improve the manuscript, and we include the revised version with this comment. We give detailed responses to specific comments below. For reviewers 1 and 2, the responses are only slightly expanded versions of the original replies to comments. For reviewer 3, we include a new detailed response here, which we did not have time to complete before the close of open discussion. Reviewer comments are in bold, and responses in normal font. The revised manuscript and differences document are attached as a supplement.

**Reviewer 1**

**p2 L28: what does "steady state form of a landscape" mean here? You've just convinced me it doesn't exist in these settings...this is a bit more clear that you mean something like a flux steady state after reading the rest of the paper, but it seems that there is no steady landscape form except in the vertical-contacts case.**

We changed the wording in this sentence, and added an additional sentence to clarify that we are talking about a flux steady state rather than topographic steady state.

**p3 L19 A change in process (e.g. away from stream-power erosion) under steep conditions breaks this relationship, as noted above on L10 or so. This is discussed to some extent but could bear more emphasis. These boundaries are the very places where erosion processes are changing. For example, some of the same authors have published on how blocky debris from strong lithologies locally alters the erosion by streams in these settings. The change to effectively a transport-limited system may necessitate at least a change in the exponents, if not the form, of the erosion law. It is clear from the later discussion that the authors appreciate this; it would be useful at this point perhaps to point out that the formulation in Eq. 3 is effectively a reference case, deviations from which may reflect the process variability present in any particular landscape.**

We agree with the reviewer on this point and have added a couple of sentences to make this assumption explicit.

**p4 L15 What is considered "subhorizontal" here? How close to horizontal can**

**the contact be before this singularity becomes important? It is rare in nature (but common in LEMs) to have a perfectly uniform, mathematically horizontal dip over a significant distance. I suggest adding an extra set of lines (or two) to Fig. 3 with some dip cases close to horizontal, perhaps 5 and 10 dip, in addition to the vertical and pure horizontal cases.**

Subhorizontal is defined on Lines 1-2 of page 4. It is whenever rock dip is small compared to channel slope. Therefore, the cases shown actually span a wide range of possible contact and channel slopes. There is not a simple way that we can think of to show specific other choices of dip angle. We have modified the main text and figure caption to make it clearer that these two limits are not explicitly a function of rock dip, but rather a comparison between rock dip and channel slope.

**p4 L18 "solely a function of erodibility." In this framework. I would argue that process variation is critical here. There is certainly field support for a retreat rate that is independent of slope but a function of drainage area in relevant landscapes, a la Crosby and Whipple 2006 (cited) and Berlin and Anderson 2007 JGR (not cited but quite relevant). But another way to view this singularity is that perhaps n=1 works well away from contacts in sub-horizontal rocks but the stream power erosion law itself is not a good model in these situations. As noted, this is also where numerical inaccuracies may become very important in LEMs. I appreciate the authors pointing out where numerical models may diverge from reality when considering this continuity framework.**

We agree. For $n = 1$ the horizontal retreat rate is a function of erodibility AND drainage area and independent of slope. This is a direct consequence of stream power erosion law. (In chi space, for $n = 1$ the horizontal retreat rate is a function of erodibility and independent of slope AND recharge area.) We have corrected the text to include

drainage area as a factor influencing retreat rate. We also agree that the singularity precludes validity of the stream power erosion law in these situations because it causes the predicted slopes to not be small enough. We have also added a few lines to the discussion concerning the cited field work and implications for the n=1 case.

**p6 L13 “time-averaged incision rate through both rock types...” This needs some clarification. Do you mean vertical incision rate in both rocks is identical to the uplift rate? That doesn’t seem quite right. Averaged over what time period?**

**p6 L17-18 “continuity state is a type of flux steady state” Here this is presented as if it follows from the above analysis, but it was stated on line 13 above that the analysis is based on assuming flux steady state. It reads as being a circular argument, but perhaps the phrasing just needs some clarification.**

This section was not very clear and did appear circular. We have edited it to make it clearer. From the results of the simulations, and specifically the fact that the landscape is periodic in chi space, you can argue that the system must be in a flux steady state. Using this conclusion, we can derive full profiles. Finally, that the whole story holds together is further confirmed by the fact that we can match the simulated profiles using the equation derived from flux steady state.

**p7 L27 “two cycles through the rock layers” not clear what this means - what cycles? The perturbation has traversed two sets of contacts?**

Not exactly - it means the knickpoint caused by the perturbation has travelled so far upstream that two sets of contacts now separate it from the downstream end of the channel that is being perturbed. The number of contacts it traversed on its way (if any) depends on the ratio between horizontal retreat rates and knickpoint celerity.

As a side note, knickpoints pass from one lithology to another unobstructed. They get damped through formation of stretch zones as in Royden and Perron (2013) and through interfering with one another. We have edited the text to try to clarify this point.

**p7 L30 how does layer thickness affect this result? Presumably it affects the distances across which a profile is developed in each rock type. A common geological scenario is thinner layers of hard rock between thick layers of soft rock. Will thin layers of hard rock slow down knickpoints for less time than thick ones, reducing the damping lengthscale? The analytical expressions and 1D modeling here stick to equal thicknesses of each type. I suspect the general result is the same, but pointing out the effect would be useful, and how to account for it in the framework described on p7. I see this issue is addressed to some extent in the 2D model setup, but its effect is not then discussed, and the 200 and 300 m alternating thicknesses are similar enough that I wouldn't expect a big impact. What about 100 m of weak rock alternating with 10 m strong-rock interbeds?**

Only the thickness of the stronger layer influences this length scale. This results because the problem is asymmetric with respect to the two rocks. The strong rock knickpoints are always slower. The time for the weak knickpoint to catch up depends on only three things: 1) how big of a head start the strong knickpoint has, 2) the velocity of the weak knickpoint, 3) the velocity of the strong knickpoint. The velocities of the two knickpoints are independent of layer thickness. The head start of the strong knickpoint is only dependent on the thickness of the strong rock. Therefore, the thinner the strong rock layer, the quicker the knickpoints should decay.

We agree that it would be interesting to simulate some cases with thin, hard layers, both to test the predictions of our theory for unequal thickness, and because this is a common situation in nature. We now include new simulation results with thin

strong layers and have expanded the discussion to examine the implications of layer thickness on steady state form.

**p10 L4-5 It's pretty hard to call the reach corresponding to a caprock waterfall a "channel", especially once flow is detached from the face. I think eSurf gives you the space to elaborate a bit more on how processes might commonly change in these settings (see my notes above) and how in general one would incorporate this into the continuity framework (without detailed exploration of such a case).**

We agree that processes dramatically change in this setting. Our speculation in the manuscript is that stream power erosion, specifically in subhorizontal rocks with $n<1$ is one possible mechanism to drive the system toward the caprock waterfall state. Once the system reaches this state, stream power erosion has certainly broken down. We have slightly expanded this part of the discussion, and make it clearer that the material on caprock waterfalls is a speculation about a possibility rather than something we have definitively shown.

**Minor notes p2 L17: "responce" change to "response" Fig 7 caption is missing punctuation at the end.**

These typos were corrected.

**Reviewer 2**

**This manuscript on the influence of horizontally (or close to it) layered rocks and**

**their influence on landscape evolution is very interesting. Some of the results are extremely counter-intuitive, and that always makes for a fun read. The math and modeling seem sound to me, and I'm generally supportive of this paper. The paper is timely, as another paper on a similar topic recently came out – Forte et al., which is cited here. Forte et al., also discussed that steady state is not reached with horizontal layers. Where this paper falls a bit short, in my opinion, is a lack of much discussion and also some lack in details of the modeling. As for the discussion, I thought they might tie in more with the Forte paper at some point, but that never happened. But in general I did not find the discussion to be very deep. As for the modeling, it was not always clear to me why the models were set-up as they were.**

**My general comment is just to give a bit more detail, including around the figures, and some suggestions for this are laid out in my line-by-line comments.**

We thank the reviewer for their careful reading of the manuscript, and pointing out a number of items that were unclear or deserved further elaboration. Detailed responses to comments are given below.

**Line by line comments: After reading the abstract I'm still not sure what channel continuity means. Notably, the sentence starting on line 5 made no sense to me, and I think that made me stumble through the rest of the abstract. I went back and read it after reading the manuscript and then it made sense to me. I think it was hard for me to envision what retreat in the direction parallel to a contact meant without the schematics, but after seeing the schematics it seems obvious. I don't have a great suggestion for improving this sentence.**

We agree that it is difficult to understand without a figure. We have expanded this part

of the abstract in an attempt to more carefully explain what we mean by continuity. The abstract is now a bit long, but we hope it is clearer.

**The caption in Figure 2 and main text around it confuse me. In A, is the upper layer steeper, or is it simply that the upper layer is overhanging the lower layer, creating an instability? Similarly, in B, isn't the problem that there was a dam created? Equation 2: Is this vertical incision rate?**

We have attemped to make this clearer. In case A, the upper layer can become steeper or create an overhang, it depends on the sign of the dip of the contact. In case B, the lower layer can create a dam or a low slope zone, depending on the sign of the dip. We have adjusted the text accordingly. Equation 2 does contain the vertical erosion rate, which we now specify in the text.

**Page 4, first paragraph. I see the math, but this is confusing. A few things. I wonder if it would be helpful to remind people the relationship between $K_w$ and $K_s$? As for equation 5 with n<1, the prediction is so counter to my 'gut', that I wonder if some discussion about whether n<1 is realistic, or about whether this counter intuitive relationship has been observed, would be useful. The n=1 case is also difficult for me to wrap my head around. Maybe more discussion is coming later.**

It is definitely counter to the common intuition based on prior work. One of the main points of this manuscript is that the assumption behind prior work actually can break down in subhorizontal rocks. We try to explain this in lines 6-8 of this page. Basically, it results because horizontal retreat rate (or knickpoint celerity) is lower for steeper channels in the case where n<1. This is because, for the same rate of vertical erosion,

horizontal retreat rates are less for steeper slopes. In cases where n<1, the increased erosion from the channel becoming steeper isn't sufficient to offset the slope effect. At n=1, these two effects are totally balanced, so slope has no effect on horizontal retreat rate. We have expanded this section to try to make it a bit clearer, and have reminded the reader that $K_w > K_s$. We do also discuss later cases where n<1 might be reasonable.

**Page 4, line 28: What does it mean that experiments with resolution suggest that the conclusions are not affected by numerics? Does that mean you changed the resolution and ran with different numerical schemes, and got the same answer? Or that your results are not dependent on the resolution for a given implementation of stream power? Please clarify.**

Our original statement was a bit too vague. We have edited this to clarify that we ran some higher resolution simulations for some cases that produced the same result.

**Page 5, line 6, 7: Is layer thickness thought to vary with uplift? I don't think so. Why do you do this?**

We are specifically examining cases where there are many different rock layers, such that the influence of base level perturbations dies out and we can see the continuity equilibrium form. This was just a practical way of generating a similar number of contacts in both the high and low uplift cases (albeit both with parameter ranges that are within the range of natural landscapes).

**Page 5, L 14: What do you mean it holds if 'slope is replaced with slope'?**

Slope is replaced with "slope in $\chi$-elevation space." We agree that the repeated word obscures the meaning a bit. We now use the word "steepness" instead, and then note in a parenthetical that by steepness we mean slope in $\chi$-elevation space.

**Figure 4 caption: What is meant by the 'steady state profile predicted by the theory'? Just the elev-chi plot for a channel with that erodibility in vertical layers? Or is it the theory that you present in this paper. I'm confused.**

We mean the theory presented in this paper. We have edited the caption to clarify this and have added the relevant equation references.

**Page 7, summary in paragraph on line 25: I got a bit lost. I think a bit more description/hand holding for the reader would help. I recognize that $\lambda^*$ is a way to show how large $\chi_{s,+}$ is. But in the description with respect to figure 6, the damping is described in terms of cycles through rock layers. I don't understand what this means, or how to get that from the equations. I must be missing something easy. How does $\chi_{s,+}$ related to the depth of the rock layers? How do I know from $\lambda^*$ how many layers the knickpoint has propagated through?**

$\chi_{s,0}$ is the $\chi$ length of the strong layer reach near base level at the moment that the weak layer becomes exposed at base level. Consequently, this distance is less than the profile distance spanned by a pair of weak and strong rocks, but is also on the same order of magnitude. The dimensionless damping length scale, $\lambda^* = \lambda/\chi_{s,0}$, therefore provides a rough (conservative) estimate of the number of strong/weak pairs that the knickpoint will pass before significant damping. We have expanded this text to clarify this point.

**Figure 7 is difficult for me to interpret. I think I can see the knickpoint that is propagating up in elevation, but I can't really make out the knickpoint that it is 'catching'. Can you tell us how you determined that there was a knickpoint at the red line that was caught? If I look at the dashed line (intermediate time) in C, it does not look like there is any significant change in the chi-elev relationship at the red line, but I think that there is supposed to be a knickpoint there, right? Or at least one close to it that will soon catch up? I only see one knickpoint downstream from there, but maybe I am interpreting incorrectly? Actually, after watching the movies, I may understand this. But I still think it is worthwhile to point out to readers exactly what you are calling knickpoints.**

Figure 7 was the best way we could think of to show this statically, though it is much clearer in the animations, as we state in the text. We think that the point of confusion here is that the knickpoints we are talking about are just sudden changes in slope, and can correspond to increases or decreases in slope (depending on $n$). We have expanded this explanation in the text.

**Fastscape runs: It is a bit unsatisfying that the n=2/3, 3/2 runs have channels that extend through 4+ layers of each rock type, but the n=1 run only just barely taps three week layers. I know this is a lot to ask, but it'd be more satisfying to see more of the n=1 profiles, i.e. just make the K values in this run smaller. I'm not adamant about this, as the 2D runs appear to be very similar to the 1D runs.**

We agree that the choice of parameters for the $n = 1$ case was suboptimal. We have rerun the simulation with a lower K value, and now the simulation has a similar number of weak and strong layers.

**In the beginning of Section 4, the authors mention that they include hillslope processes. It seems like this needs a bit more description. How are the different rock types treated with the hillslope model? How do they model hillslopes?**

We are using the standard hillslope diffusion approach employed in Fastscape, which does not have the capability to adjust the diffusion coefficient with rock type. We now clarify this in the text.

**Discussion and Conclusions: I liked that the authors brought in a real world example. However, this example confused me. I may be wrong, but my impression of Niagara Falls and the Niagara river is that the soft rocks underneath the hard caprock are indeed basically vertical at the waterfalls. But if you move any length downstream the channel is not so steep anymore. I might be wrong as I haven't studied the Niagara River, just visited it. But does the whole length of profile have the 'inverted' relationship (steeper in weak rocks) suggested by Figure 4D, or is it just 'inverted' around the waterfall? This may seem a picky point, but I would guess, as the authors brought up elsewhere, that the processes going on right at the knickpoint are not adequately modeled by stream power. So in some ways this comparison feels a bit odd to me. I felt as though the discussion could be expanded a bit.**

If the stream power erosion law continued to hold as the channel steepened, then in theory one would expect the entire channel to remain steep in the weak rocks (for $n<1$)). However, the stream power erosion law breaks down for such steep channels as erosion processes take over that are not well-described by the stream power law. Therefore, we are only speculating that continuity can push the system toward this state (by first making the channel steep in the weak rocks). From a more standard assumption of topographic equilibrium, one would never expect steepening in weak

rocks, so it is not clear how you would approach such a state to begin with. There are potentially other explanations, such as non-locality in erosion processes near the contact, that cannot be entirely ruled out. We have expanded this discussion slightly and tried to make it clearer that it is speculative. In the specific case of Niagara Falls, which is one of the most famous of many possible examples, the flattening below the waterfall in part occurs because of the nearby base level imposed by Lake Ontario.

**The parameter n turns out to be extremely important in this study. Any thoughts beyond Niagara Falls on how your contribution plays in to the n debate? Have many studies suggested that n<1? Are there any other landscapes to call upon to illustrate the modeled behavior besides Niagara Falls? I also generally prefer a separate conclusions section. I think it is better for authors because often times the only sections of the paper that get read are the abstract and conclusions. But this is stylistic.**

We do not attempt to constrain what realistic values of n should be. There are theoretical arguments that some incision processes will produce n<1 (e.g. Whipple et al. [2000] cited in this work, or Covington et al. [2015], GRL). While there are likely good field sites where a natural experiment could be used to test the ideas developed here, we think that finding and studying such a site is beyond the scope of this manuscript. Our main goal here is to solidify our theoretical understanding of the equilibrium behavior of the stream power erosion law in layered rocks.

We have subtantially expanded the discussion and written a separate conclusions section. The expanded discussion begins with a comparison to the work of Forte et al. (2016), and uses this as a framework to discuss implications that were not fully developed in the previous version. We have also tried to clarify our ideas concerning caprock waterfalls.

**Reviewer 3 (Whipple)**

**This manuscript is well written and entails an important step forward in understanding the influence of rock strength variations in landscape evolution. The novel focus is on the influence of the slope exponent ($n$) in the stream power river incision model on landscape evolution in areas where sub-horizontal layered rocks with varying rock strength are exposed – extending beyond a recent treatment from my group (Forte et al., 2016, Earth Surface Processes and Landforms) that considered only the n = 1 case. It is remarkable that the venerable stream power model still holds surprises! Though of course it is always important to consider the degree to which processes and effects not encapsulated in the stream power model will alter the behavior of natural landscapes.**

**There is much value in the analysis and discussion presented. Reading and carefully reviewing this paper has notably advanced my own understanding of how landscapes described by the stream power model will evolve in the presence of layered rocks as a function of the relative strength between stronger and weaker layers, the relative thickness of strong and weak layers, and the dip of the contacts (only simple planar dip panels considered thus far) in cases with $n < 1$ or $n > 1$. As part of the process of reviewing this paper I re-derived most of the key relationships and updated an existing 1d finite-difference solver to handle a series of dipping layers with variable erodibility (K in the stream power model) and variable thickness so I could test both the author's initially counter-intuitive results (such as the formation of cliffs in the weak units, not the strong units, if $n < 1$) and my own derivations. I find complete agreement with Figures 3, 4, 5, and 8. Similarly, though I would word some aspects differently (reflecting differences in my derivations described below), I agree with the points made in the discussion and conclusions. Thus I agree with all the findings in**

**a qualitative sense. Likewise I see no problems with the numerical simulation results – both in 1d and 2d using FastScape.**

We thank the reviewer for this thorough review that has helped us to better understand the results that we present in this manuscript and to rethink and expand aspects of our approach.

**However, I do not agree with some of the derivations and prefer a different approach to solving the problems discussed and explaining the interesting results of the 1d profile evolution models. As the only way I felt I could evaluate the derivations was to redo them following my own intuition for how to pose the problem, I present alternative solutions below. Rather than working the derivations here, I outline the logic the present the solution. Hopefully this will prove an effective and constructive approach. The alternate derivation given below results in an identical solution for horizontal bedding (Eqn 5), which is good, but suggests differing sensitivities to the dip of contacts and the relative thicknesses of strong and weak units.**

Our disagreement centers around the general approach and conceptual model used to explain the observed behavior of the stream power model. Here we summarize our understanding of these differences, and provide a general response. More detailed responses to individual points are included below. Our approach uses a concept we called continuity. Continuity is a natural generalization of topographic equilibrium (where erosion rates are constant everywhere, with steepness adjusting to rock strength to accommodate those equal erosion rates). We define continuity at a contact as a condition where erosion rates in both rock types are equal in the direction parallel to the contact surface. In the case of vertical contacts, this produces equal erosion as is found in the case of topographic equilibrium. Continuity can also be applied along

an entire profile. We argue that negative feedback between topography and erosion will tend to drive the system toward a state of continuity, in analogy to the negative feedback that results in topographic equilibrium. However, rather than assumming that profiles will approach a state of continuity, we have used this as a seemingly reasonable hypothesis to test against simulation results. In Section 2, we now more carefully explain that erosional continuity is a hypothesis to be tested.

Whipple did not find our approach using continuity intuitive, which perhaps means that it can be improved or at least more clearly stated. He begins by examining knickpoint celerity in the different rock layers. Knickpoint celerity is mathematically identical to horizontal retreat rate. For n not equal to 1, he suggests that either the weak (if $n > 1$) or strong (if $n < 1$) layer controls the horizontal retreat rate of the contact (i.e. knickpoint celerity). The controlling layer maintains the steepness that it would have if it were the only rock layer and were in equilibrium with uplift rate. The non-controlling layer adjusts its horizontal retreat rate (knickpoint celerity) to match the retreat of the controlling layer. Since celerity and horizontal retreat are identical, the above conceptual framework is the same as our statement of continuity in the case of horizontal rocks. If the dip of the contact is non-zero, then the two approaches differ, in that the celerity approach is matching horizontal retreat rates at the contact, and the continuity approach is matching retreat rates in the direction parallel to the contact.

We may be missing something here, but we do not see an a priori reason why one rock layer should control the other, why this control should differ in cases with $n < 1$ and $n > 1$, or why explicitly celerity, rather than retreat rate, should be matched at the contact. On the last point, it is at least clear that celerities do not match in the case of vertical contacts, which is why the proposed celerity approach only applies to the horizontal limit. However, whether or not the proposed set of assumptions can be justified in advance, they do provide testable predictions about profile shapes that can be compared against the simulations. These predictions are also different than those produced by the continuity model, enabling a comparison of the two possible

conceptual approaches with the numerical results.

The easiest prediction to test from the proposed celerity model is that the controlling rock layer maintains an equilibrium slope. If this were true, then it would certainly invalidate our conclusion that a flux steady state is reached, where long-term average vertical incision rates at any x position in the profile are equal to the uplift rate. This invalidation would result from the fact that incision rates in the controlling layer are equal to uplift. If that were true, then in order to match average incision to uplift, the incision rates in the non-controlling layer would also have to be equal to uplift. However, as pointed out by both us and Forte et al. (2016), the vertical incision rates are not equal in the two rock layers.

The above reasoning, along with our conclusion in the manuscript that the simulated profiles are in a flux steady state, leads us to believe that the proposed celerity model is not precisely correct. However, it can also be tested more explicitly using simulation results. For $n < 1$, the controlling rock layer is hypothesized to be the strong layer. If we examine a snapshot from a simultion just as the base level encounters a weak rock layer, then we can clearly see the equilibrium slope produced at base level within the strong rock (where uplift and erosion are equal). This slope can be compared with the slope attained far from base level, and they are not, in general, the same (Fig. 1). In his response to our short comment, Whipple also notes he observed some disagreement between his model and the slopes observed in the $n < 1$ case.

For the $n > 1$ case, the controlling rock layer is the weak layer. We can again compare the slope at base level with the slope observed far from base level (Fig 2). Here the case is not as clear, in part because of oscillations in slope that are produced by the base level perturbations. The first weak layer above the base level layer actually overshoots the continuity equilibrium slope (and goes to even lower slopes). Layers further up oscillate back toward the slope observed at base level, but ultimately settle on a slope that is slightly less than that at base level (see upper two weak layers). This oscillation is more easily observed in the animations. Since the difference in slope is

not large, it is not surprising that Whipple comes to the conclusion that the simulations satisfactorily confirm his hypothesis. However, this is in part a result of the parameter choice and can also be predicted by our theoretical relations. Using the continuity theory, one can explicitly predict the ratio of continuity steady state slope ($S_{1,cont}$) to the slope at base level in a given rock layer ($S_{1,topo}$). Combining equations 1, 5, and 8 gives

$$\frac{S_1, cont}{S_{1,topo}} = \left( \frac{H_1/H_2 + (K_1/K_2)^{1/(1-n)}}{1 + H_1/H_2} \right)^{1/n} \tag{1}$$

This relation is depicted (Fig. 3) as a function of $n$ for $H_1 = H_2$ and $K_w = 2K_s$. ( *Note: Since this figure and equation actually make an interesting and useful point, and help to clarify the difference between the two types of equilibrium, we have added this to the manuscript along with some additional discussion.*) For $n < 1$, the ratio of continuity steady state slope to topographic steady state slope is always substantially different from one, for both the strong and weak rocks, which is why the difference is more visible in these simulations. For $n > 1$, the adjustment of the weak rock, which is suggested to be the controlling layer, is always less than a factor of 2. For our simulated value of $n = 1.5$ it is only a difference of 10-20%. The continuity theory predicts slightly more difference in slope as n approaches 1. We simulate a case with $n = 1.2$ and, as predicted, observe a slightly larger difference between base level and far from base level slopes in the weak rock (Fig 4).

To summarize our current conclusions about cases with $n \neq 1$, we can see why one might come to the conclusion that one rock layer is controlling the retreat of the other, and derive a slope relationship based on knickpoint celerity matching. In fact, our theory predicts this to be an approximate solution for a variety of parameter choices (Fig 3). However, we do not think that this celerity model provides a precise description of the far from base level equilibrium state, as we have demonstrated by comparison with simulations. Rather than one rock layer controlling the other, the celerities of both rock layers adjust to meet in the middle. Is is true, however, that in most cases one rock

layer is adjusting more than the other. We think that the approximate success of the knickpoint celerity model in horizontal rocks results exactly because knickpoint celerity is identical to horizontal retreat rate. The more general statement of continuity matches the retreat rate in the direction of the contact. As a result, the theory works for arbitrary dip, not just in the horizontal limit.

For $n = 1$, we think that the model presented by Whipple is essentially correct. Horizontal retreat rate is independent of slope, and therefore adjustments in slope cannot produce a match in horizontal retreat in the two rock layers. We think that this should ultimately lead to a state where the entire profile is retreating at the rate of the weak rock (at least if overhangs are not allowed to form). We have not tested whether this is precisely true in the numerical simulations, but we suspect that the retreat rate in simulations may be sensitive to the discretization scheme, because of numerical dispersion in the vicinity of the sharp features that form in the profile. Though the numerical schemes typically used in landscape evolution models enforce continuity at the contact, the continuity theory breaks down for the rest of the profile when $n = 1$ in horizontal rocks. Since continuity breaks down, we were having difficulty understanding the $n = 1$ case, and here we think that Whipple's explanation makes dynamics of $n = 1$ channels much clearer. We have adjusted our discussion accordingly.

**First, I don't much like the conceptual model in Figures 1 and 2. Most important, a problem only arises in the strong-over-weak case: overhangs cannot be sustained, as illustrated in Figure 2a. Conversely, as illustrated by Forte et al. (2016) and commonly seen in nature, weak rocks can readily be stripped off the top of strong rocks, leaving a tapering wedge of weak rock in the case of an upstream-dipping contact like that shown in Figure 2b. I also don't like the use of the word "continuity" for this, since in much of the geomorphic and fluid flow literature "continuity" means conservation of mass, though I appreciate that**

**you are imposing a continuous profile with no overhangs.**

We do not think that continuity, as we defined it, applies only to the case of strong-over-weak rock. We think that negative feedback, as described by these figures, will in general tend to drive the system toward continuity. We only consider this line of reasoning to be suggestive that continuity is a reasonable hypothesis, and we test this hypothesis against the simulations. Relationships derived from continuity do predict stream profile shapes for stream power erosion with $n \neq 1$. In the case of $n = 1$, continuity cannot be preserved. In the simulations it is preserved at the contact because of the numerical scheme (e.g. overhangs are not allowed because the channel can only have one z position for every x position) rather than for physical reasons.

In our simulations where $n \neq 1$ we do not see stripping occur when weak layers are on the top (see animations in supp. material). We do, however, see this kind of stripping for $n = 1$, as noted by Forte et al. (2016). We agree that this form of plateau topped with weak layers is commonly found in nature, which could be an argument for $n = 1$ in those cases (or perhaps for the inapplicability of the stream power model in the steep topography). Alternatively, we do often see a flat plateau preserved at the top of the topography that results from the initial condition of flat topography. Our previous simulations had weak layers on top, but we have run additional cases with strong layers on top. This plateau is preserved independent of which layer is on top. There is some dependence on $n$, with stonger preservation of the flat initial condition when $n > 1$.

We agree that the choice of the word "continuity" is somewhat unfortunate. We were never completely satisfied with this word, though we also had not thought of the conflict with the more common meaning of "continuity" associated with conservation relationships. So far, we have not been able to think of a more appropriate word, though we are open to any suggestions. We will change it to "erosional continuity" to try to distinguish it from conservation of mass.

**I find it most useful to think about this problem in terms of the controls on the kinematic wave speed that characterizes the evolution of river profiles governed by the stream power incision model (Rosenbloom and Anderson, 1994): $Ce = KA^mS^{(n-1)}$. Key elements are (1) all else equal the kinematic wave speed is higher in weak rocks than strong, and (2) wave speed decreases with Slope for $n < 1$, is independent of Slope for $n = 1$, and increases with Slope for $n > 1$. The surprising results in this paper all stem from the curious effect that wave speed decreases with Slope for $n < 1$.**

Kinematic wave speed happens to be numerically equal to horizontal retreat rate, so we agree with these statements when horizontal rocks are considred. While it may be more intuitive to think about celerity, we do not see a way to satisfactorily predict the simulation results from a celerity based approach. If such an approach could be found that successfully predicted the simulation results, and was more intuitive to other workers in the field, we would certainly consider using it instead. However, after some thought, we have not yet found such an approach.

**From study of the evolution of 1d river profiles cutting through layered rocks for cases $n < 1$, $n > 1$, and $n = 1$ revealed in numerical simulations (as in Figures 4 and 5), I suggest below a set of fundamental controls on the development of profile shape (cliffs formed in the weak rock ($n < 1$), the strong rock ($n > 1$), or through each strong-over-weak couplet ($n = 1$)), and the retreat rate of the slope-break knickpoint at the strong-over-weak contact.**

**The authors come close to stating what I believe is happening in the case of horizontal contacts: (1) fundamentally cliffs are forming because all-else held equal the kinematic wave speed of profile segments within the weak unit exceeds that of segments within the strong unit, so there is a tendency to undermine, or to form consuming knickpoints at strong-over-weak contacts, but as described**

**by the authors and illustrated in the numerical simulations, the river profile will evolve toward an equilibrium where the upstream migration rate of the strong unit matches that of the weak unit at the contact; (2) for $n < 1$ wave speed decreases with increasing slope, so in response to the tendency to undermine, the profile steepens in the weak unit until the wave speed of the weak unit at the contact has slowed to equal the wave speed of the strong unit at the contact (the strong unit maintains an equilibrium slope at the contact and the knickpoint at the contact migrates at the rate set by the wave speed of the strong unit on an equilibrium slope – this best describes the basal strong-over-weak contact, see below); (3) for n=1 wave speed is independent of slope, so the river profile has no way to respond, and a vertical cliff forms (50% in weak, 50% in strong unit) with retreat rate = wave speed of the weak unit – the cliff grows in height until the full strong-over-weak doublet is incorporated and the retreat rate is set by the wave speed of the weak unit; (4) for $n > 1$ wave speed increases with increasing slope, so in response to the tendency to undermine, the profile steepens in the strong unit until the wave speed of the strong unit at the contact has increased to equal the wave speed of the weak unit at the contact (the weak unit maintains an equilibrium slope at the contact and the knickpoint at the contact migrates at the rate set by the weak unit on an equilibrium slope).**

**Once this realization is made, it is easy to write equations for the wave speed in each unit at the contact, set them equal, and solve for the ratio of the slope of the weak unit ($S_w$) to the slope of the strong unit ($S_s$). For horizontal beds, Equation 5 in the paper is recovered. So the derivation given is exact in the limit of horizontal contacts (also satisfies expectation for vertical contacts). However, in my analysis the derivation for the case of non-horizontal contacts (Eqn 2) appears to be incorrect. First, the solution only applies for strong-over-weak scenarios, so E1, S1 (downstream) could only be the weak unit. Second, if the derivation described above for horizontal contacts is generalized to account for planar dipping beds, a different solution is found.**

**In the case of dipping beds, the migration rate of the knickpoint at the strong-over weak contact is not set only by the kinematic wave speed ($Ce = E/S = KA^m S^{(n-1)}$) as it is for horizontal contacts, but must account for the slope of the contact ($S_c$). For example, if the contact were exactly parallel to the river bed, then the migration rate would approach infinity over that reach. Geometrically one can readily show that the local knickpoint migration velocity ($Ce_{kp}$) will be: $Ce_{kp} = E/(S - S_c)$ (which correctly reduces to the kinematic wave speed for $S_c = 0$).**

**(Note here that the caption to Fig. A1 indicates that Sc in the paper is defined positive for an upstream dip, while channel slope S is positive downstream. I worry that this could prove very confusing. Here I instead define Sc as positive for downstream dip).**

We have changed this sign convention to avoid confusion.

**As noted above, the migration rate of the slope-break knickpoint at the contact is set by the equilibrium wave speed within the strong unit for $n < 1$ (at least for the basal strong-over-weak contact), and within the weak unit for $n > 1$ - the problem then is how the dip of the contact amplifies or reduces knickpoint velocity relative to the kinematic wave speed. Solving for the equivalent of Eqn 5 in the presence of dipping beds, I find (derivations available on request):**

$$S_w/S_s = (K_w/K_s)^{(1/(1-n))} * (1 - S_c/S_s)^{(1/(1-n))}; for\, n < 1 \quad (2)$$

**and**

$$S_w/S_s = (K_w/K_s)^{(1/(1-n))} * (1 - S_c/S_w)^{(1/(n-1))}; for\, n > 1. \quad (3)$$

**Note that $S_c/S_s$ appears in the $n < 1$ case because the retreat rate is set by the wave speed in the strong unit, and $S_c/S_w$ appears in the $n > 1$ case because the retreat rate is set by the wave speed in the weak unit.**

ESurfD
**These solutions, however, only obtain over a range of $S_c/S_s \lesssim 1$ or $S_c/S_w \lesssim 1$ - basically restricted to sub-horizontal conditions - as outlined below. In addition, as mentioned above, the $n < 1$ solution applies best to the basal strong-over-weak contact: the oversteepening of the weak unit is damped up-section because the slope-break knickpoints at the strong-over-weak contacts act as a local baselevel, reducing local incision rate within the overlying strong unit (as happens in weak-over-strong contacts with $n = 1$). This causes a decrease in the slope within the strong unit, which increases the kinematic wave speed and thus decreases the degree of over-steepening of the weak unit. This complicating phenomenon is restricted to the $n < 1$ case.**

**I have tested these revised equations, and the limits on their applicability outlined below, against numerical simulations with satisfying results.**

Above we outlined both why we think this solution does not adequately explain the simulation results, as well as why it should provide an approximate solution in some parts of the parameter space (e.g. $n \gg 1$), which is actually predicted by the continuity theory.

**For $n < 1$ and downstream-dipping beds ($S_c$ is positive), the solution only applies for $S_c/S_s < 1 - -(K_s/K_w)(1/n)$: for larger (more positive) downstream dips, an equilibrium profile results ($S_s$ and $S_w$ have equilibrium values equal to the vertical contact case even though knickpoints are slowly migrating upstream over time). For $n < 1$ and upstreamdipping beds ($S_c$ is negative), preliminary comparison with numerical simulations indicates the solution is only valid for $|(S_c/S_s)| \lesssim 1$. For steeper upstream dips, the profile transitions toward an equilibrium form (I have not studied this in detail).**

**For $n > 1$ and downstream-dipping beds ($S_c$ is positive), the solution only**

**applies for $Sc < Sw$. At $Sc = Sw$, knickpoint velocity is infinite. For $Sc > Sw$, the strong-over-weak contact propagates downstream, invalidating the analysis. For $n > 1$ and upstream-dipping beds ($S_c$ is negative), the solution only applies for $S_c/S_w > 1 - (K_w/K_s)^{(1/n)}$: for larger (more negative) upstream dips, an equilibrium profile results ($S_s$ and $S_w$ have equilibrium values equal to the vertical contact case even though knickpoints are slowly migrating upstream over time).**

We would like to point out that in addition to working in the subhorizontal case, for $abs(S_c/S_s) \gtrsim 1$, our Eqn 3 correctly predicts slopes similar to topographic equilibrium slopes for steep or vertical dips.

**These solutions can be re-cast into the form of Eqn 2 (note I have inverted the relation here):**

$$E_2/E_1 = E_s/E_w = (S_s - S_c)/S_w; for\, n < 1 \quad (4)$$

$$E_2/E_1 = E_s/E_w = S_s/(S_w - S_c); for\, n > 1 \quad (5)$$

**Thus Eqn 2 should have two forms, one for $n < 1$ and one for $n > 1$. (remember that $S_c$ is defined here as positive downstream).**

**Section 3.2. I did not attempt to reproduce or critically evaluate Equation 8, but found no dependence of erosion rate patterns on $H_1/H_2$ in my numerical simulations. For horizontal beds, Equation 5 is exactly satisfied for a very wide range of $H_1/H_2$. I did not investigate whether a greater sensitivity to layer thickness emerges with dipping contacts.**

Equation 5 only predicts slope ratios, and we agree that it is independent of the thickness ratio ($H_1/H_2$). However, the absolute slopes, and consequently erosion rates, do depend on this relative thickness. Actually, this is an interesting prediction

of our model that we had not fully tested or explored. In the revised version of the manuscript we present plots that compare simulations with non-equal thicknesses to some of our previous simulations. Keeping uplift, erodibility, and $n$ the same, an increase in the percentage of the thickness that is occupied by weak layers results in a decrease in the slopes in both weak and strong units. In the limit where one rock layer is much thicker than the other, then this thick layer has erosion rates that match uplift, and it has a slope that would be the same as its slope in the case of topographic equilibrium. This also has interesting implications for natural systems, where thicknesses will not typically be regular. We have expanded the discussion to consider these results.

**Section 3.3. I don't see the profile as being "perturbed" at baselevel because, as the authors note on page 6, line 27, new river segments formed at baselevel always begin in equilibrium ($E = U$, and equilibrium slopes). The perturbations develop above as the differential wave speeds near contacts begin to manifest in deviations from equilibrium slopes and erosion rates. Thus I'm not enthused about the "damping length scale" terminology. However, the result appears robust – differential wave speeds are rapidly accommodated at the first strong-over-weak contact, with knickpoints at contacts quickly converging on a migration velocity set by the equilibrium wave speed of either the weak unit ($n >= 1$) or the strong unit ($n < 1$).**

We think this is a language issue: perturbed from which equilibrium? The newly forming segment does have an erosion rate equal to uplift. Segments at distances from base level that are farther than the damping length scale have erosion rates that depend only on the rock type, are not equal to uplift rate, and are collectively in flux steady state. That is what we mean by "damping length scale" (the scale over which erosion rates converge toward continuity steady state). Given that the manuscript

is about continuity steady state, and the base level reach that is at topographic steady state produces disturbances that travel up the profile and decrease in size as they go, we think that it is reasonable to consider these as perturbations that are damped. This is certainly the visual impression we get when watching animations of the simulations. However, we have also edited the text throughout to try to make it clear which equilibrium we are referring to at any given time.

**That said, I am confused by Eqn 10. First, there appears to be a typo in Eqn (10): as derived, the last term should be $A_0(m/n)$ not $A(m/n)$. Further, Ce = kinematic celerity = horizontal migration rate of river "patch" (patch as used by Royden and Perron, 2012) = $KAmS(n-1)$. For a steady-state river patch, $S = (U/K)(1/n)A(-m/n)$. Combining these, Whipple and Tucker (1999) showed that the horizontal migration rate of a steadystate river patch is $Ce = U(n-1)/nK(1/n)A(m/n)$ – this is the relation given for Eqn 10, so the equation appears to be correct, but the derivation (and the apparent typo) implies it is incorrect.**

There was a typo in this equation, $A$ should have been $A_0$, as noted. The celerity we are deriving is also celerity in $\chi$ space, which may have been another source of confusion. We now clarify this in the text.

**Finally, although widely appreciated, it seems worth stating that readers should beware the difference between the mathematics of the stream power model (SPM), insightful though they can be, and the physical reality of nature. Many processes are not represented in the SPM and therefore predictions may fail. Despite this, I am very supportive of publishing papers like this that explore model predictions because this allows one to: (1) generate testable hypotheses, constrain parameters, or recognize where models fail and why; (2) use any**

**failures to improve the model; and (3) know what will happen in landscape evolution simulations based on the SPM under different conditions.**

We are aware of the limitations of SPM and hope that we express this clearly in the manuscript. One of the central points of the manuscript is to clarify that common (implict) assumptions about equilibrium landscapes do not always hold for subhorizontal layered rocks. So, to the extent that we are using the SPM and these assumptions to interpret real landscapes, we should be aware of these limitations. However, beyond this, we provide a framework that can extend the standard topographic equilibrium model and account for subhorizontal rocks. Whether evidence of continuity steady state is to be seen in nature is still uncertain. However, our theoretical work does generate testable hypotheses, and help us to understand what is occuring in landscape evolution simulations.

**I have a few additional comments listed below with reference to page and line number.**

**1. Title: I suggest revising title to remove "continuity" as this will mean "conservation of mass" to many. Also I suggest emphasizing your key finding about the dependence on n, if you can find an effective wording.**

We have changed this to "erosional continuity" to try to clarify this difference. We have not been able to think of a better word to describe this concept than "continuity."

**2. Page 1, Line 21-22: This is not true. Many studies of bedrock channel morphology are expressly seeking information about the history of climate, tectonics, or drainage divide migration recorded in non-steady state profiles (as you note on page 2, line 4).**

Here our text was unclear. Of course bedrock channel profiles are often used to explore transience, but an understanding of steady state is used to identify the transience. We have edited these sentences to try to make our meaning clearer.

**3. Page 2, line 9: better to not call the stream power model (SPM) a "law".**

We agree and have modified the text accordingly.

**4. Page 2, line 11-12: the SPM is widely used in modeling studies, but is not required as a basis of profile analysis – channel steepness and concavity can be measured and interpreted in terms of relative uplift rate, climate, or rock strengths independent of the river incision rule.**

We agree that profile analysis doesn't require specific use of a river incision rule, so we have edited this sentence to remove the "stream power model" phrase.

**5. Page 3, line 3-4: as you show in your analysis, this is not true for $n<1$.**

The wording here was not precisely correct. We have edited it to note that we are considering retreat rates specifically in the direction of the contact plane and vertical erosion rates. We also now make it clearer that the idea of continuity is treated as a hypothesis, which is tested using the simulations.

**6. Page 3, line 5-6: I disagree. Where a weak layer overlies a strong layer, there is no constraint on the relative stream segment migration speed – the weak**

**layer can be stripped off, leaving a bench on the underlying strong layer or a tapering wedge of the weak layer. Such forms are very common in nature.**

We do think that creation of a low or reverse slope segment will reduce erosion rates as a result of reduced slope in the contact zone. However, as discussed above, we treat this as a hypothesis to be tested by simulations. The models do seem to display this behavior, except for $n = 1$.

**7. Page 9, line 9: This sentence is confusing since channel segments formed at baselevel are always initially at equilibrium (E=U, steady state form) in systems described by the SPM.**

It is true that these segments are in topographic equilibrium at baselevel. Topographic equilibrium assumes conditions stay constant. So if conditions, including K, change with time / uplift, topographic equilibrium is not actually an equilibrium for the system. Our system has an equilibrium different from topographic, so we can (and successfully do) treat the events at the baselevel as perturbations. However, we agree that it is potentially confusing to use the phrase “disequilibrium at base level,” and we have slightly reworded to clarify our meaning.

**8. Page 9, line 16: “channel steepness” here would be better written as “channel slope” or “channel gradient”, since “steepness” commonly refers to the steepness index.**

Agreed.

Please also note the supplement to this comment:
http://www.earth-surf-dynam-discuss.net/esurf-2016-41/esurf-2016-41-AC3-supplement.zip
* * *
[Figure]

**Fig. 1.** High uplift simulation with n=2/3. The slope attained in the strong rock far from base level is different than the topographic steady state slope that is attained near base level. Grey is weak rocks.

[Figure]

**Fig. 2.** High uplift simulation with n=1.5. The slope attained in the weak rock far from base level is different than the topographic steady state slope that is attained near base level. Grey is weak rocks.

[Figure]

**Fig. 3.** Contrast between continuity steady state and topographic steady state slopes as a function of n, with equal layer thickness and Kw = 2 Ks.

[Figure]

**Fig. 4.** High uplift simulation with n=1.2 The weak rock has stronger slope contrasts than in the n=1.5 case, as predicted by erosional continuity.

---

## Author Response (AR2)

Dear editor,

  Since the comments by the editor did not suggest any specific revisions, we have completed an additional reading and revision of the manuscript to check for any changes that we desired to make. We have made several small revisions to clarify points or correct typos. These are noted in this marked up manuscript.

Sincerely,
Matt Covington

[revised manuscript text omitted]